# Structural Estimation of Partially Observed Linear Non-Gaussian Acyclic Model: A Practical Approach with Identifiability

**Songyao Jin[•], Feng Xie[◇], Guangyi Chen[‡•], Biwei Huang[⋆], Zhengming Chen[†•], Xinshuai Dong[‡], Kun Zhang[‡•]**

[•]Mohamed bin Zayed University of Artificial Intelligence, Abu Dhabi, UAE
[◇]Beijing Technology and Business University, Beijing, China
[⋆]University of California San Diego, La Jolla CA, USA
[†]Guangdong University of Technology, Guangzhou, China
[‡]Carnegie Mellon University, Pittsburgh PA, USA

## Abstract

Conventional causal discovery approaches, which seek to uncover causal relationships among measured variables, are typically sensitive to the presence of latent variables. While various methods have been developed to address this confounding issue, they often rely on strong assumptions about the underlying causal structure. In this paper, we consider a general scenario where measured and latent variables collectively form a partially observed causally sufficient linear system and latent variables may be anywhere in the causal structure. We theoretically show that with the aid of high-order statistics, the causal graph is (almost) fully identifiable if, roughly speaking, each latent set has a sufficient number of pure children, which can be either latent or measured. Naturally, LiNGAM, a model without latent variables, is encompassed as a special case. Based on the identification theorem, we develop a principled algorithm to identify the causal graph by testing for statistical independence involving only measured variables in specific manners. Experimental results show that our method effectively recovers the causal structure, even when latent variables are influenced by measured variables.

## 1 Introduction

Inferring causal relationships from observed data is a popular field within statistics and artificial intelligence. Conventional causal discovery methods, such as the PC algorithm (Spirtes & Glymour, 1991), Greedy Equivalence Search (Chickering, 2002), LiNGAM (Shimizu et al., 2006), are designed to identify causal structures among measured (observed) variables, assuming that no latent variables exist in the causal graph. However, in practice, usually we can not measure all task-related variables, leading to a violation of the no-latent-variable assumption (Spirtes & Zhang, 2016).

To estimate causal relationships among measured variables involving latent variables, one line of approaches assumes independent latent confounders and utilizes conditional independence constraints (Spirtes et al., 1995; Colombo et al., 2012; Claassen et al., 2013), or functional causal models and non-Gaussianity (Hoyer et al., 2008; Chen & Chan, 2013; Tashiro et al., 2014; Maeda & Shimizu, 2020). Going beyond this assumption, some approaches allow for causally related latent confounders. These include methods based on measurement models and hierarchical models, distinguished by whether the latent variable's children are measured or not. In particular, measurement model-based methods typically rely on the Tetrad condition (Silva et al., 2006; Kummerfeld & Ramsey, 2016), non-Gaussianity (Shimizu et al., 2009; Cai et al., 2019; Xie et al., 2020), mixture oracles (Kivva et al., 2021). On the other hand, hierarchical model-based methods have evolved from tree-like structures (Pearl, 1988; Zhang, 2004; Choi et al., 2011; Drton et al., 2017) to more general hierarchical structures (Xie et al., 2022; Huang et al., 2022; Kong et al., 2023). Of particular relevance to us are methods based on the Generalized Independent Noise (GIN) condition (Xie et al., 2020; 2022), which leverage higher-order statistics (beyond the second-order moments in statistics,

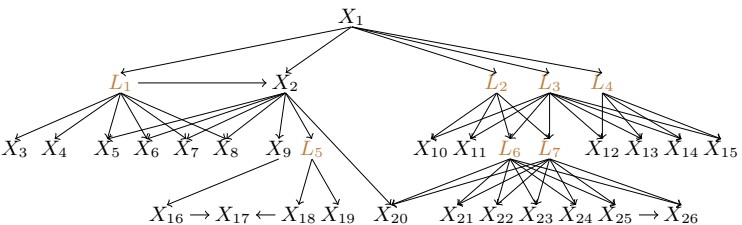

| | |
|---|---|
| V: A variable | |
| **V**: A set of variables | |
| $\mathcal{U}$: An atomic unit | |
| $\mathbb{U}$: A set of atomic units | |
| $\|\mathbb{U}\|$: Number of variables | |
| PCh: Pure children | |
| MS: Measured surrogates | |

Figure 1: An example latent causal graph $\mathcal{G}$ that our algorithm can handle. $L_i$ are latent variables, and $X_i$ are measured variables.

Table 1: Graph notations. More Details in Appdix. I.

e.g., skewness, kurtosis, etc. of the data) to improve identifiability in the presence of latent confounders. However, all those methods typically rest on the *measurement assumption*–that there are no measured variables acting as parents of any latent variables in the underlying causal graph.

The aforementioned methods are tailored to specific configurations among measured or latent variables, which limits their practical applicability. In fields like human health, latent genetic factors, for instance, might be influenced by observed environmental factors, subsequently affecting observed host phenotypes. While Adams et al. (2021) established necessary and sufficient conditions for structure identifiability based on overcomplete ICA, it does not have a practical estimation approach and requires knowing the number of latent variables at the beginning. Recently, Squires et al. (2022) presented a method that allows latent variables to be influenced by measured variables, but it still relies on strong assumptions, such as the absence of edges between pairs of observed or latent nodes.

In this paper, we target an inclusive scenario involving latent variables, without imposing constraints on their presence and positions. Meanwhile, we seek to strike a delicate balance between theoretical identifiability and practical feasibility. To identify the causal structure within this general scenario, we leverage the linear causal model and non-Gaussianity, giving rise to the proposed Partially Observed Linear Non-Gaussian Acyclic Model (PO-LiNGAM). We demonstrate that, even without prior knowledge of the latent variables, the causal graph is virtually identifiable when each variable conforms to the PO-LiNGAM and each latent set has a sufficient number of pure children. Building upon the identification theorem, we introduce a feasible search algorithm that utilizes the Generalized Independent Noise (GIN) condition in specific manners. The proposed algorithm comprises three iteratively executed phases that sequentially identify latent variables and infer causal relationships among (both latent and measured) variables, progressing from leaf to root variables. The algorithm halts its inference of the higher-level causal structure when the latent set lacks a sufficient number of pure children. This helps to maintain precision and applicability in real-world situations. Furthermore, in cases where there are no latent confounders, our algorithm's results will reduce to those obtained using LiNGAM-based methods (Shimizu et al., 2006; 2011). Additionally, our algorithm possesses the capability to handle overlapping variables between discovered latent sets, an aspect often overlooked by other methods. Our contributions are summarised as:

a.) We consider a broad spectrum of causal structures, regardless of the presence and positions of latent variables, and establish their theoretical identifiability under milder graphical conditions. Figure 1 depicts an illustrated latent graph with several notable characteristics. For instance, the latent variable set $\{L_2, L_3\}$ serves not only as hidden confounder for measured variables but also for latent variables, forming a latent hierarchical structure. Additionally, $\{L_3, L_4\}$ has its own unique set of pure children, distinct from those of $\{L_2, L_3\}$, though both are influenced by the measured variable $\{X_1\}$. Latent triangles, such as $\{X_1, L_1, X_2\}$, are accommodated as well.

b.) We propose an efficient algorithm to estimate the causal structure through a precise application of GIN tests, without prior knowledge of latent variables. Our algorithm infers the causal graph by precisely identifying the pure and impure children.

## 2 PARTIALLY OBSERVED LINEAR NON-GAUSSIAN ACYCLIC MODEL

### 2.1 MODEL DEFINITION

Precisely, we focus on a partially observed linear non-Gaussian acyclic model where $\mathbf{X} = \{X_1, \ldots, X_m\}$ denotes the set of measured variables, $\mathbf{L} = \{L_1, \ldots, L_n\}$ denotes the set of latent variables, and $\mathbf{V} = \mathbf{X} \cup \mathbf{L}$ denotes the total set of variables. Each variable $V_i \in \mathbf{V}$ is generated

within a directed acyclic graph (DAG) by the following linear structural equation model:

$$\mathrm{V}_i = \sum_{\mathrm{V}_j \in \mathrm{Pa}(\mathrm{V}_i)} b_{ij} \mathrm{V}_j + \varepsilon_{\mathrm{V}_i} \qquad (1)$$

where $\mathrm{V}_i$ and $\mathrm{V}_j$ can be either measured or latent variables, $\mathrm{Pa}(\mathrm{V}_i)$ is the set of parent variables of $\mathrm{V}_i$, $b_{ij}$ represents the causal strength from $\mathrm{V}_j$ to $\mathrm{V}_i$, and $\varepsilon_{\mathrm{V}_i}$ represents the noise term of $\mathrm{V}_i$. The noise terms are assumed to be continuous random variables with non-Gaussian distribution, and they are independent of each other. Additionally, without loss of generality, we assume that all variables have zero mean (Otherwise, they can be centered).

**Definition 1** (Partially Observed Linear Non-Gaussian Acyclic Model (PO-LiNGAM)). *A linear graphical model, with its directed acyclic graph $\mathcal{G}$, is called a partially observed linear non-Gaussian acyclic model, iff:*

- *Each variable in variable set $\mathbf{V}$ is generated by the structural equation model (1).*
- *The distribution over $\mathbf{V}$ is Markov and faithful to the DAG $\mathcal{G}$.*

Unlike the LiNGAM in Shimizu et al. (2006), we allow for the presence of latent variables. As shown in Definition 1, we disregard the specific positions of the measured and latent variables. This means that both measured and latent variables, individually or jointly, can act as parents to other measured or latent variables. It distinguishes our work from previous studies on linear latent models (Silva et al., 2006; Hoyer et al., 2008; Cai et al., 2019; Xie et al., 2020; 2022; Squires et al., 2022; Huang et al., 2022). In this paper, our goal is to establish the identifiability of the PO-LiNGAM causal structure and show how it can be discovered solely from the set of measured variable $\mathbf{X}$.

## 2.2 GRAPHICAL CONDITION FOR IDENTIFIABILITY: FORMULATION

We now formulate the graphical condition that will be utilized to establish the structural identifiability in Section 3.4. Below, we first give some definitions corresponding to graphical structures.

**Definition 2** (Pure Child). *Let $\mathbf{V}$ be the set of all variables that have the common parent variable set $\mathbf{P}$ and $\mathbf{V} \cap \mathbf{P} = \emptyset$. For any $\mathrm{V}$ in $\mathbf{V}$, $\mathrm{V}$ is a pure child of $\mathbf{P}$ iff $\mathrm{V}$ is d-separated from $\mathbf{V} \backslash \mathrm{V}$ given the parents $\mathbf{P}$. Similarly, for any subset $\mathcal{V}$ of $\mathbf{V}$, $\mathcal{V}$ is a pure child of $\mathbf{P}$ iff $\mathcal{V}$ is d-separated from $\mathbf{V} \backslash \mathcal{V}$ given the parents $\mathbf{P}$.*

**Example 1**. *In Figure 1, $\{X_5\}, \{X_6\}, \{X_7\}, \{X_8\}$ are pure children of $\{L_1, X_2\}$, respectively or in combination. $\{L_1, X_2\}$ in combination is a pure child of $\{X_1\}$.*

A complex causal graph can be identified by decomposing it into simpler, independent components. This simplifies the analysis by focusing on individual components and their relationships. We now give the definition of the atomic unit which represents the basic building block of causal structures.

**Definition 3** (Atomic Unit). *An atomic unit $\mathcal{V}$ is either a single measured variable, or a set of variables $\mathbf{V}$ with size $\|\mathbf{V}\| = t \geq 1$ that the following conditions hold:*

- *$\mathbf{V}$ contains at least one latent variable.*
- *$\mathbf{V}$ has at least $t + 1$ atomic units as its pure children, in which in total at least $t$ child variables are contained in the atomic units other than $t$ atomic units, and each of all these pure atomic unit children[1] covers all atomic units that share common variables with it. Additionally, when $t \geq 2$, $\mathbf{V}$ has another atomic unit that either neighbors it or overlaps with it.*
- *$\mathbf{V}$ is the minimal set that cannot be decomposed into several smaller atomic units.*

**Example 2**. *In Figure 1, each measured variable is an atomic unit. $\{L_6, L_7\}$ is an atomic unit with size $t = 2$, where in addition to the 2 (note that $2 \geq t$) pure atomic unit children $\{X_{21}\}$ and $\{X_{22}\}$, the remaining pure atomic unit children $\{X_{23}\}$ and $\{X_{24}\}$ in total contain 2 child variables. Furthermore, $\{L_2, L_3\}$ is an atomic unit with size $t = 2$, where in addition to the 2 pure atomic unit children $\{X_{10}\}$ and $\{X_{11}\}$, the remaining pure atomic unit child $\{L_6, L_7\}$ contain 2 child variables. However, $\{L_1, X_2\}$ is not strictly an atomic unit because it can be decomposed into two smaller atomic units $\{L_1\}$ and $\{X_2\}$.*

For measured variables, it is reasonable to examine each variable individually. As for latent variables, linear transitivity allows us to extract their information provided they have a sufficient number

---

[1]It indicates that the pure children themselves are atomic units. The same goes for the following.

of pure children. Below, for the sake of convenience, we default to considering children at the atomic unit level. It means that when we refer to $\mathcal{C}$ as a pure child of $\mathcal{P}$, the premise is that $\mathcal{C}$ and $\mathcal{P}$ are atomic units. Below is the graphical condition to guarantee identifiability of PO-LiNGAM.

**PO-LiNGAM Identifiability Condition** : *a.) Any latent variable is in at least one atomic unit. b.) For any two atomic units with overlapping variables, their non-overlapping parts do not influence each other.*

The first term is the basic condition for identification of latent variables (Xie et al., 2020; Huang et al., 2022). It implies that every latent variable must have measured variables as its descendants. The second term arises because the discovered atomic units may have overlapping variables. In cases where two atomic units share variables, we cannot determine whether the non-overlapping part of one unit causally affects the non-overlapping part of another unit due to insufficient information. Theorem 4 will show the entire causal graph is identifiable at the atomic unit level when the above graphical (identifiability) condition is satisfied.

## 3 Structure Identification through a Three-Phase Algorithm

In this section, we present our algorithm for structural identification and corresponding theoretical guarantees. Our algorithm is iterative, with each iteration consisting of three phases, progressively uncovering the entire causal graph from leaf to root nodes. Let $\mathbb{A}$ be the *active atomic unit set* consisting of atomic units under investigation, which is initially set to $\mathbf{X}$ and updated progressively. Phase I identifies the causal relationships among atomic units in $\mathbb{A}$. Phase II discovers new atomic units from $\mathbb{A}$. Phase III refines the discovered atomic units. After that, $\mathbb{A}$ is fed back to phase I, and a new iteration starts until $\mathbb{A}$ is not updated. Finally, we simply verify multiple root atomic units by checking the discovered second-highest causal order atomic units (see Appendix C.5).

---

**Algorithm 1:** The identification procedure of PO-LiNGAM

---

**Input:** Measured variable set $\mathbf{X}$
**Output:** Causal graph $\mathcal{G}$
1 Initialize partial causal graph $\mathcal{G} := \emptyset$, active atomic unit set $\mathbb{A} := \mathbf{X}$;
2 **while** $\|\mathbb{A}\| > 1$ **and** $\mathbb{A}$ *is updated* **do**
3     $(\mathcal{G}, \mathbb{A}) \leftarrow$ IdentifyLeafNodesAndTheirParents$(\mathcal{G}, \mathbb{A})$; // Phase 1
4     $(\mathcal{G}, \mathbb{A}, \text{NewAtomicUnitSet}) \leftarrow$ DiscoverNewAtomicUnits$(\mathcal{G}, \mathbb{A})$; // Phase 2
5     $(\mathcal{G}, \mathbb{A}) \leftarrow$ RefineAtomicUnits$(\mathcal{G}, \mathbb{A}, \text{NewAtomicUnitSet})$; // Phase 3
6 **end**
7 Check for multiple root atomic units (see Appendix C.5);
8 **return** $\mathcal{G}$

---

### 3.1 Phase I: Identifying leaf nodes and their parents

Each iteration starts with phase I. In phase I, we deal with the identification of causal structure among atomic units in $\mathbb{A}$ by recursively identifying the leaf nodes in the causal graph composed of atomic units in $\mathbb{A}$ and their latent confounders. Below, we briefly review the Generalized Independent Noise (GIN) condition (Xie et al., 2020), which is the workhorse for our algorithm.

**Definition 4** (GIN condition) (Xie et al., 2020). *Let $\mathbf{Z}$ and $\mathbf{Y}$ be sets of variables with zero mean in a linear non-Gaussian acyclic causal model. We say that $(\mathbf{Z}, \mathbf{Y})$ follows* GIN *condition if and only if $\omega^{\mathrm{T}} \mathbf{Y}$ is independent of $\mathbf{Z}$, where $\omega$ satisfies $\omega^{\mathrm{T}} \mathbb{E}[\mathbf{Y}\mathbf{Z}^{\mathrm{T}}] = 0$ and $\omega \neq 0$.*

Roughly speaking, if the GIN condition is satisfied, then there is a set of variable $\mathbf{P}$ (which can contain variables in $\mathbf{Y}$) of size $\|\mathbf{P}\| < \|\mathbf{Y}\|$, which are ascendants of $\mathbf{Y}$ and d-separate $\mathbf{Y} \backslash \mathbf{P}$ from $\mathbf{Z} \backslash \mathbf{P}$. See Xie et al. (2020) and Appendix F for more details.

In a causal graph, for any leaf node variable, it is d-separated from the other variables given its parents. By utilizing the GIN condition and non-Gaussianity, this graphical condition can be statistically linked to the data. To derive the theorem for the general case, we start by considering a scenario in which all variables are measured, meaning there are no latent variables in the causal graph–this corresponds to the case of LiNGAM (Shimizu et al., 2006; 2011).

**Proposition 1** (Identifying leaf variables without latent confounders). *Let* $\mathbf{V}$ *be a set of variables satisfying PO-LiNGAM with no latent variables, and* V *be a variable in* $\mathbf{V}$*. For a set of variable* $\mathbf{P} \subseteq \mathbf{V}\backslash V$, V *is one of the current leaf variable nodes and* $\mathbf{P}$ *contains all parent variables of* V, *iff* $(\mathbf{V}\backslash V, V \cup \mathbf{P})$ *follows the* GIN *condition and there is no* $\tilde{\mathbf{P}} \subset \mathbf{P}$ *such that* $(\mathbf{V}\backslash V, V \cup \tilde{\mathbf{P}})$ *follows the* GIN *condition.*

**Example 3**. *Consider the causal graph in Figure 1 composed of* $\mathbf{V} = \{X_9, X_{16}, X_{17}, X_{18}\}$*. Let* $V = X_{17}$ *and the minimal* $\mathbf{P} = \{X_{16}, X_{18}\}$*.* $(\{X_9, X_{16}, X_{18}\}, \{X_{17}, X_{16}, X_{18}\})$ *follows the* GIN *condition, which implies that* $X_{17}$ *is the current leaf node, and* $X_{16}$ *and* $X_{18}$ *are its parents. Removing* $X_{17}$ *from* $\mathbf{V}$*, we can find that* $X_{16}$ *is the current leaf node and* $X_9$ *is its parent.*

If there are no latent variables, Proposition 1 alone suffices to identify the entire causal graph. We achieve this by recursively identifying the leaf variables and then reconsidering the remaining variables. In practice, however, it is difficult to exhaustively measure and collect all task-related variables (Spirtes et al., 2000). Thanks to the transitivity of linear causal relations, although we cannot directly access to the latent variables, we can use their measured descendants to test GIN conditions.

**Definition 5** (Surrogate variable & Measured Surrogate Variable set $\mathrm{MS}_{a,b}$). *Let* $\mathcal{P}$ *be an atomic unit involving latent variables and atomic unit* $\mathcal{C} \in \mathrm{PCh}(\mathcal{P})$*. Surrogate variables of* $\mathcal{P}$ *are the variables in each* $\mathcal{C}$ *or up to* $\|\mathcal{C}\|$ *surrogate variables of* $\mathcal{C}$*. Measured Surrogate variables of* $\mathcal{P}$ *are its surrogate variables that are measured. Measured Surrogate Variable set* $\mathrm{MS}_a$ & $\mathrm{MS}_b$ *of* $\mathcal{P}$ *are two sets of measured surrogate variables, where the minimal set d-separating* $\mathrm{MS}_a$ *and* $\mathrm{MS}_b$ *is* $\mathcal{P}$*, and* $\|\mathrm{MS}_a(\mathcal{P})\| = \|\mathrm{MS}_b(\mathcal{P})\| = \|\mathcal{P}\|$*.*

**Example 4**. *In Figure 1, for the atomic unit* $\{L_6, L_7\}$*,* $\mathrm{MS}_a$ *can be* $\{X_{21}, X_{22}\}$ *and* $\mathrm{MS}_b$ *can be* $\{X_{23}, X_{24}\}$*. For the atomic unit* $\{L_2, L_3\}$*,* $\mathrm{MS}_a$ *can be* $\{X_{21}, X_{22}\}$ *and* $\mathrm{MS}_b$ *can be* $\{X_{10}, X_{11}\}$*.*

To harmonize the view, in the following sections, for each measured variable that also serves as an atomic unit, we designate both $\mathrm{MS}_a$ and $\mathrm{MS}_b$ to refer to the measured variable itself. We will explain in Phase II how to discover latent atomic units and their measured descendants. Let us now show how the leaf atomic units and their parent atomic units in the causal graph can be identified by utilizing measured surrogate variable sets of each known (measured or latent) atomic unit.

**Theorem 1** (Identifying leaf atomic units). *Let* $\mathbb{U}$ *be a set of non-overlapping atomic units satisfying PO-LiNGAM with known* $\mathrm{MS}_{a,b}$ *for each atomic unit in* $\mathbb{U}$*, and* $\mathcal{U}$ *be an atomic unit in* $\mathbb{U}$*. For a set of atomic units* $\mathbb{P} \subseteq \mathbb{U}\backslash\mathcal{U}$*,* $\mathcal{U}$ *is one of the current leaf atomic unit nodes and* $\mathbb{P}$ *contains all parent atomic units of* $\mathcal{U}$*, iff* $(\mathrm{MS}_a(\mathbb{U}\backslash\mathcal{U})^2, \mathrm{MS}_b(\mathcal{U} \cup \mathbb{P}))$ *follows the* GIN *condition and there is no* $\tilde{\mathbb{P}} \subset \mathbb{P}$ *such that* $(\mathrm{MS}_a(\mathbb{U}\backslash\mathcal{U}), \mathrm{MS}_b(\mathcal{U} \cup \tilde{\mathbb{P}}))$ *follows the the* GIN *condition.*

**Example 5**. *Consider the causal graph in Figure 1 composed of* $\mathbb{U} = \{\{X_2\}, \{X_9\}, \{L_5\}\}$*. Suppose we know that* $\{L_5\}$ *is the latent confounder of* $\{X_{18}\}$ *and* $\{X_{19}\}$*, meaning that* $\mathrm{MS}_{a,b}(\{L_5\})$ *can be* $\{X_{18}\}$ *and* $\{X_{19}\}$*, respectively.* $(\{X_2, X_9\}, \{X_2, X_{19}\})$ *follows the* GIN *condition, which implies that* $\{L_5\}$ *is the current leaf node and* $\{X_2\}$ *is its parent that d-separates it from the other atomic units. Removing* $\{L_5\}$ *from* $\mathbb{U}$*, we can find that* $\{X_9\}$ *is another leaf node and* $\{X_2\}$ *is its parent.*

Theorem 1 extends Proposition 1, allowing for the atomic units involving latent variables. In Theorem 1, we make the assumption that each atomic unit does not overlap with other atomic units. However, as shown in Figure 1, $\{L_2, L_3\}$ and $\{L_3, L_4\}$ are two atomic units with the overlapping variable $L_3$. In our algorithm, during each iteration, Phase III checks for overlap among atomic units after the new atomic units are discovered in Phase II. Below, we propose the complement to Theorem 1 that ensures the theorem's correctness even in the presence of overlapping atomic units starting from the second iteration onwards.

**Remark 1** (Complement to Theorem 1). *In Theorem 1, before testing* GIN *condition, a.) if any atomic units in* $\mathbb{U}$ *have overlapping variables with* $\mathcal{U}$*, then we remove them from* $\mathbb{U}$ *and remove them from* $\mathbb{P}$ *if they exist in* $\mathbb{P}$*. b.) if the atomic units in* $\mathbb{P}$ *overlap with each other, instead of using entire set of* $\mathrm{MS}_b(\mathbb{P})$ *for testing, we use its maximal subset* $\mathbf{S}$ *that makes* $(\mathrm{MS}_a(\mathbb{U}\backslash\mathcal{U}), \mathbf{S})$ *does not follow* GIN *condition. Similarly, for* $\tilde{\mathbb{P}}$*, instead of using entire* $\mathrm{MS}_b(\tilde{\mathbb{P}})$ *for testing, we use its maximal subset* $\tilde{\mathbf{S}}$ *that makes* $(\mathrm{MS}_a(\mathbb{U}\backslash\mathcal{U}), \tilde{\mathbf{S}})$ *does not follow* GIN *condition.*

---

² $\mathrm{MS}_a(\mathbb{U}\backslash\mathcal{U})$ denotes the union of $\mathrm{MS}_a$ of each atomic unit in $\mathbb{U}\backslash\mathcal{U}$. The same goes for the following.

We now can infer the causal structure among atomic units in $\mathbb{A}$. Specifically, the atomic units in $\mathbb{A}$ and their latent confounders form a causal graph. For any leaf atomic unit of the graph in $\mathbb{A}$, if all its parent atomic units are also in $\mathbb{A}$, it and its parents can be identified by using Theorem 1 and Remark 1. Then, we archive it and remove it from $\mathbb{A}$. The remaining atomic units in $\mathbb{A}$ and their latent confounders form a smaller causal graph in which we can continue identifying leaf atomic units. Below, we present the main search procedure of Phase I. The algorithm details can be found in Algorithm 2 (Appendix C.1), which also includes an additional checking and refinement process (lines 8 and 14, Alg. 2) that will be explained in the next subsection.

**Phase I**: Identifying Leaf Nodes And Their Parents

1. Start with **partial causal graph** $\mathcal{G}$ and **active atomic unit set** $\mathbb{A}$.
2. For each atomic unit $\mathcal{U}$ in $\mathbb{A}$, test if there is a subset of atomic units $\mathbb{P}$ in the $\mathbb{A}\backslash\mathcal{U}$ that satisfies **Theorem 1** with **Remark 1**.
3. If found, add direct edges from each atomic unit in $\mathbb{P}$ to $\mathcal{U}$, and remove $\mathcal{U}$ from $\mathbb{A}$.
4. Repeat steps 2 and 3 until no more atomic units can be removed from $\mathbb{A}$.
5. Return $\mathcal{G}$ and $\mathbb{A}$.

**Example 6** [An illustration of Phase I]. *Consider the causal structure in Figure 1. Suppose that at the beginning, the active variable set is $\mathbb{A} = \{\{X_1\},\ldots,\{X_{26}\}\}$. With Phase I, one can know that* $\mathrm{Pa}(\{X_{17}\}) = \{\{X_{16}\},\{X_{18}\}\}$, $\mathrm{Pa}(\{X_{16}\}) = \{X_9\}$, *and* $\mathrm{Pa}(\{X_9\}) = \{X_2\}$.

### 3.2 PHASE II: DISCOVERING NEW ATOMIC UNITS

After there are no leaf atomic units with their full parents in current $\mathbb{A}$, which can be identified by Phase I, we discover new atomic units by clustering their pure children from $\mathbb{A}$ in Phase II. This procedure is achieved by first clustering the child atomic units that have common parents, and then identifying pure children to discover new atomic units.

**Theorem 2** (Clustering atomic units). *Let $\mathbb{U}$ be a set of non-overlapping atomic units satisfying PO-LiNGAM with known $\mathrm{MS}_{a,b}$ for each atomic unit in $\mathbb{U}$, and there is no such leaf atomic unit in $\mathbb{U}$ that its full parents are also in $\mathbb{U}$. Let $\mathbf{Y} \subseteq \mathrm{MS}_b(\mathbb{Y})$ be a part of measured surrogate variables of $\mathbb{Y}$, where $\mathbb{Y}$ is a subset of $\mathbb{U}$ and no $\tilde{\mathbb{Y}} \subset \mathbb{Y}$ satisfies $\mathbf{Y} \subseteq \mathrm{MS}_b(\tilde{\mathbb{Y}})$. The atomic unit set $\mathbb{Y}$ have a total of $\|\mathbf{Y}\| - 1$ parent variables (including latent variables), denoted as $\mathbf{P}$, which can d-separate $\mathbb{Y}$ from $\mathbb{U}\backslash(\mathbb{Y}\cup\mathbf{P})$, if a.) $(\mathrm{MS}_a(\mathbb{U}\backslash\mathbb{Y}), \mathbf{Y})$ follows the GIN condition, b.) there is no subset $\tilde{\mathbf{Y}} \subset \mathbf{Y}$ such that $(\mathrm{MS}_a(\mathbb{U}\backslash\mathbb{Y}), \tilde{\mathbf{Y}})$ follows the GIN condition, and c.) there is no any atomic unit $\mathcal{P} \in \mathbb{Y}$ such that $(\mathrm{MS}_a(\mathcal{P} \cup \mathbb{U}\backslash\mathbb{Y}), \mathbf{Y})$ follows the GIN condition.*

**Example 7**. *Consider the causal graph in Figure 1 composed of $\mathbb{U} = \{\{X_2\},\{X_3\},\ldots,\{X_8\}\}$. We can find five $\mathbb{Y}$ clusters that satisfy Theorem 2, i.e., $\mathbb{Y}_1 = \{\{X_3\},\{X_4\}\}$, $\mathbb{Y}_2 = \{\{X_5\},\{X_6\},\{X_7\}\}$, $\mathbb{Y}_3 = \{\{X_5\},\{X_6\},\{X_8\}\}$, $\mathbb{Y}_4 = \{\{X_5\},\{X_7\},\{X_8\}\}$, and $\mathbb{Y}_5 = \{\{X_6\},\{X_7\},\{X_8\}\}$.*

Similar to Theorem 1, Theorem 2 focus on the non-overlapping atomic units. Furthermore, for two overlapping atomic units, if one atomic unit is entirely covered by another, the parents of the covered atomic unit must be a subset of or identical to the parents of the covering atomic unit. To leverage the covering atomic unit, we introduce a complement to Theorem 2, which extends its applicability to overlapping atomic units.

**Remark 2** (Complement to Theorem 2) . *In Theorem 2, before testing GIN conditions, if any atomic units in $\mathbb{U}$ is completely covered by any one atomic unit in $\mathbb{Y}$, then we remove these covered atomic units from $\mathbb{U}$.*

Next, we propose Corollary 1. It helps identify individual pure children of set $\mathbb{Y}$'s common parents from $\mathbb{Y}$, thus providing information for the new atomic units.

**Corollary 1** (Identifying individual pure child). *Let $\mathbb{U}$ be a set of atomic units satisfying PO-LiNGAM with known $\mathrm{MS}_{a,b}$ for each atomic unit in $\mathbb{U}$, and there is no such leaf atomic unit in $\mathbb{U}$ that its full parents are also in $\mathbb{U}$. Let $\mathbb{Y}$ be a subset of $\mathbb{U}$ that satisfies Theorem 2 with Remark 2, and $\mathcal{Y}_1$ be an atomic unit in $\mathbb{Y}$. For any one atomic unit $\mathcal{Y}_2$ in $\mathbb{U}\backslash\mathbb{Y}$, iff the new subset $\{\mathcal{Y}_2 \cup \mathbb{Y}\backslash\mathcal{Y}_1\}$ also satisfies Theorem 2 with Remark 2, then $\{\mathcal{Y}_2 \cup \mathbb{Y}\backslash\mathcal{Y}_1\}$ has the same total parent set as those of $\mathbb{Y}$, and $\mathcal{Y}_1$ and $\mathcal{Y}_2$ are two individual pure children, each of which is not partially or fully covered by other atomic units.*

**Example 8**. *Continuing from Example 7, $\mathbb{Y}_2 = \{\{X_5\}, \{X_6\}, \{X_7\}\}$, $\mathbb{Y}_3 = \{\{X_5\}, \{X_6\}, \{X_8\}\}$ are two clusters that fulfill the requirements. We can know $\mathbb{Y}_2$ and $\mathbb{Y}_3$ share the same parent set because they both have $\{\{X_5\}\{X_6\}\}$, and $\{X_7\}$ and $\{X_8\}$ are two individual pure children of their common parent set because either of them can form a cluster with the base set $\{\{X_5\}\{X_6\}\}$.*

Based on the theorems above, we can discover new atomic units by identifying individual pure atomic unit children that have common parents. There is a special situation where only one common latent parent variable is behind only two atomic units, called *bi-unit cluster*. For example, $\{\{X_3\}, \{X_4\}\}$ in Figure 1. In such a situation, Corollary 1 does not help to identify whether $\{X_3\}$ and $\{X_4\}$ have direct causal relationship between them. Fortunately, Lemma 1 in Xie et al. (2022) provides a solution and is integrated into our algorithm.

Below, we present the search procedure of Phase II, where step 4 is aligned with the requirement of the second term in Definition 3. The detailed algorithm can be found in Algorithm 3 (Appendix C.2). During Phase II, we exclusively focus on pure children in current $\mathbb{A}$. Causal relationships for impure children and uncovered pure children are inferred in Phase I of subsequent iterations.

**Phase II**: Discovering New Atomic Units

1. Start with **partial causal graph** $\mathcal{G}$, **active atomic unit set** $\mathbb{A}$, and **size** $n = 1$.
2. $n = n + 1$. For each variable subset $\mathbf{Y}$ from $\mathrm{MS}_b(\mathbb{A})$ with $\|\mathbf{Y}\| = n$, test if $\mathbf{Y}$ satisfies **Theorem 2** with **Remark 2**, until all subset are tested. Collect the satisfactory $\mathbb{Y}$ where $\mathbf{Y} \subseteq \mathrm{MS}_b(\mathbb{Y})$.
3. Gather the satisfactory $\mathbb{Y}$ sets that contain some same atomic units together as a set of sets, and add it into a **collection C**, repeatedly and exhaustively. **C** is thus a collection of sets of sets.
4. For each element (set of sets) in **C**, first identify individual pure atomic unit children for the **bi-unit cluster** by **Lemma 1** in Xie et al. (2022). Then, identify individual pure atomic unit children by choosing a **base set** and testing **Corollary 1**. Additionally, with the help of identified pure children, identify whether atomic units in the **base set** are also pure children.
5. If a sufficient number of pure children that share the same parents are identified, we create new atomic units, add them into $\mathbb{A}$ and **NewAtomicUnitSet**, and remove their pure children from $\mathbb{A}$.
6. Repeat steps 2-5 until no more subsets that satisfy **Theorem 2** with **Remark 2** are found.
7. Return $\mathcal{G}$, $\mathbb{A}$, and **NewAtomicUnitSet**.

**Example 9** [An illustration of Phase II]. *Continuing from Example 6 and consider the structure in Figure 1. The current $\mathbb{A} = \{\{X_1\}, \ldots, \{X_8\}, \{X_{10}\}, \ldots, \{X_{15}\}, \{X_{18}\}, \ldots, \{X_{26}\}\}$. With Phase II, we can discover and introduce new atomic unit $\{L_1'\}$ to be parent of $\{\{X_3\}, \{X_4\}\}$, $\{L_2', L_3'\}$ to be parent of $\{\{X_5\}, \{X_6\}, \{X_7\}, \{X_8\}\}$, $\{L_4', L_5'\}$ to be parent of $\{\{X_{12}\}, \{X_{13}\}, \{X_{14}\}, \{X_{15}\}\}$, $\{L_6'\}$ to be parent of $\{\{X_{18}\}, \{X_{19}\}\}$, and $\{L_7', L_8'\}$ to be parent of $\{\{X_{21}\}, \{X_{22}\}, \{X_{23}\}, \{X_{24}\}\}$. We add the new discovered atomic units into $\mathbb{A}$ and remove their individual pure children from $\mathbb{A}$.*

Due to the mechanism of Theorem 2 and the searching procedure of Phase II, a clustering issue may arise when the parent atomic unit is fully covered by others. For example, in Figure 1, if $\{X_3\}$ was a latent atomic unit and not in current $\mathbb{A}$, then during Phase II, $\{X_4\}$ would be clustered with $\{\{X_5\}, \{X_6\}, \{X_7\}, \{X_8\}\}$. Consequently, it would be treated as sharing the same parent atomic unit $\mathcal{L}'$ that contains two variables, as other variables in the cluster. Fortunately, after $\{X_3\}$ is discovered, we can identify the sub atomic unit from $\mathcal{L}'$ by re-clustering $\{\{X_3\}, \{X_4\}\}$. We introduce Proposition 2 to check sub-atomic units in the general case and incorporate an additional procedure into Phase I (lines 8 and 14, Alg. 2). Further details can be found in Appendix C.4.

**Proposition 2** (Checking sub-atomic units) *Let $\mathbb{P}$ be a collection of a.) an atomic unit, b.) the atomic units that have common children with it, and c.) the atomic units that overlap with it. Let $\mathbb{U}$ be the set of all children of each atomic unit in $\mathbb{P}$, and $\mathbf{Y}$ be a subset of $\mathrm{MS}_a(\mathbb{Y})$ where $\mathbb{Y}$ is a subset of $\mathbb{U}$. If there is no such $\mathbf{Y}$ with $\|\mathbf{Y}\| \leq \|\mathrm{Pa}(\mathbb{Y})\|$ satisfying Theorem 2 and Remark 2 with corresponding parameters ($\mathbb{U}$, $\mathbb{Y}$, $\mathbf{Y}$), then no undiscovered sub-atomic units in $\mathbb{P}$ can be further identified.*

## 3.3 PHASE III: REFINING ATOMIC UNITS

In this phase, we check whether the newly discovered atomic units in Phase II have variable overlap with other atomic units in $\mathbb{A}$, which is relatively under-explored in other methods (Xie et al., 2020; Huang et al., 2022). Following that, we further verify each atomic unit in $\mathbb{A}$ by checking whether it can be fully decomposed into other smaller atomic units in $\mathbb{A}$.

**Theorem 3** (Checking the overlap of atomic units). *Let $\mathcal{U}_1$ and $\mathcal{U}_2$ be two atomic units. Let $\mathbf{U}_1$ be part of variables in $\mathrm{MS}_b(\mathcal{U}_1)$. Iff $(\mathrm{MS}_a(\mathcal{U}_1 \cup \mathcal{U}_2), \mathbf{U}_1 \cup \mathrm{MS}_b(\mathcal{U}_2))$ follows the GIN condition and there is no subset $\tilde{\mathbf{U}}_1 \subset \mathbf{U}_1$ such that $(\mathrm{MS}_a(\mathcal{U}_1 \cup \mathcal{U}_2), \tilde{\mathbf{U}}_1 \cup \mathrm{MS}_b(\mathcal{U}_2))$ follows the GIN condition, then the two atomic units have $\|\mathcal{U}_1\| - \|\mathbf{U}_1\| + 1$ overlapping variables in total.*

**Example 10**. *Consider two atomic unit $\{L_1\}$ and $\{L_1, X_2\}$[3] in Figure 1. Let $\mathbf{U}_1 = \{X_4\}$ and we find $(\{X_3, X_5, X_6\}, \{X_4, X_7, X_8\})$ follows the GIN condition. This implies that $\{L_1\}$ and $\{L_1, X_2\}$ have 1 = 1-1+1 overlapping variable. The same comes for $\{X_2\}$ and $\{L_1, X_2\}$.*

By Theorem 3, we identify the overlapping atomic units in $\mathbb{A}$, facilitating structure identification in the subsequent iterations. Furthermore, in the $\mathbb{A}$ after Phase II, there may exist decomposable 'atomic units' that can be decomposed and replaced by other smaller atomic units in $\mathbb{A}$. We have the following corollary to identify such situations.

**Corollary 2** (Identifying decomposable 'atomic unit'). *Let $\mathcal{U}$ be an atomic unit and $\mathbb{S}$ be a set of small atomic units that can be completely covered by $\mathcal{U}$ and do not overlap with each other. $\mathcal{U}$ can be completely decomposed as and replaced by the atomic units in $\mathbb{S}$, iff $\|\mathcal{U}\| = \|\mathbb{S}\|$.*

**Example 11**. *Continuing from Example 10, we find that both $\{L_1\}$ and $\{X_2\}$ have one overlapping variable with $\{L_1, X_2\}$ but not with each other. $\{L_1, X_2\}$ is a decomposable 'atomic unit', which can be replaced by two small atomic units $\{L_1\}$ and $\{X_2\}$.*

In our algorithm, for each identified decomposable 'atomic unit' in $\mathbb{A}$, we replace it with the small atomic units that make it up and move the children of that decomposable 'atomic unit' to them. The searching procedure for Phase III is presented below, and a detailed algorithm can be found in Algorithm 4 (Appendix C.3).

**Phase III**: Refining Atomic Units

1. Start with **partial causal graph** $\mathcal{G}$, **active atomic unit set** $\mathbb{A}$, and **NewAtomicUnitSet**.
2. For each pair of atomic units in $\mathbb{A}$ (at least one of which is also in **NewAtomicUnitSet**), test whether **Theorem 3** applies and, if so, record the number of overlapping variables.
3. For each atomic unit in $\mathbb{A}$, which overlaps with other atomic units in $\mathbb{A}$, test whether it is decomposable by **Corollary 2**. Move the children of the **decomposable 'atomic unit'** to the atomic units that decompose it, and remove it from $\mathbb{A}$.
5. Return $\mathcal{G}$ and $\mathbb{A}$.

**Example 12** [An illustration of Phase III]. *Continuing from Example 9 and consider the structure in Figure 1. With Phase III, we can find the new discovered atomic unit $\{L'_2, L'_3\}$ can be fully replaced by $\{L'_1\}$ and $\{X_2\}$. We move the children of $\{L'_2, L'_3\}$ to $\{L'_1\}$ and $\{X_2\}$, and remove $\{L'_2, L'_3\}$ from $\mathbb{A}$. After that, we start the next iteration and feed the current $\mathbb{A}$ into Phase I.*

### 3.4 IDENTIFIABILITY FOR CAUSAL STRUCTURES

In this section, we discuss the identifiability of our algorithm for the causal structure. Our algorithm runs three phases in a loop, discovering the causal graph from leaf to root nodes. The computational complexity depends on the number of variables (including latent variables) and density of underlying causal graph, which determine the number of iterations needed to discover the entire graph. A complete illustrative example and the algorithm complexity are provided in Appendix D and E, along with some experiments of running time in Appendix G.4.

**Theorem 4** (Identifiability of causal graph). *Suppose that the input data $\mathbf{X}$ follows PO-LiNGAM with the PO-LiNGAM identifiability condition. Then the atomic units, their size, and the causal structure among them can be fully identified with our algorithm. If there are no direct causal relationships between variables within atomic units, the entire causal graph $\mathcal{G}$ is virtually identifiable.*

Moreover, it is natural that under more stringent conditions, the identification of the causal graph can be accomplished using only a subset of the theorems and procedures outlined earlier.

**Corollary 3** (Identifiability of causal graph with stricter conditions). *Suppose that the input data $\mathbf{X}$ follows PO-LiNGAM. If each latent variable has at least two pure variable children, the entire causal graph can be identified with aid of Theorem 1, Theorem 2, and Corollary 1 from Phases I and*

---

[3]Strictly speaking, $\{L_1, X_2\}$ is not an atomic unit because it is decomposable. However, we did not know this when we discovered it by clustering its pure children in Phase II.

Table 2: **Comparison on synthetic data with four typical causal structures.** Results are averaged over ten experiments. "-" indicates that the method is not suitable for the corresponding graph.

| Algorithm | | Correct Ordering Rate ↑ | | | | | | Error Rate in Latent Variables ↓ | | | | | | F1-score ↑ | | | | | |
|---|---|---|---|---|---|---|---|---|---|---|---|---|---|---|---|---|---|---|---|
| | | Ours | LiNGAM | BPC | FastGIN | LaHME | IL²H | Ours | LiNGAM | BPC | FastGIN | LaHME | IL²H | Ours | LiNGAM | BPC | FastGIN | LaHME | IL²H |
| Case 1 | 5k | **0.96** | 1.0 | - | - | - | - | **0.0** | 0.0 | - | - | - | - | **0.90** | 0.94 | - | - | - | - |
| | 10k | **0.93** | 1.0 | - | - | - | - | **0.0** | 0.0 | - | - | - | - | **0.94** | 1.0 | - | - | - | - |
| | 50k | **0.96** | 1.0 | - | - | - | - | **0.0** | 0.0 | - | - | - | - | **0.99** | 1.0 | - | - | - | - |
| Case 2 | 5k | **0.96** | - | 0.28 | 0.72 | 0.70 | 0.67 | **0.05** | - | 0.0 | 0.50 | 0.55 | 0.50 | **0.98** | - | 0.71 | 0.60 | 0.70 | 0.63 |
| | 10k | **0.97** | - | 0.28 | 0.68 | 0.65 | 0.34 | **0.10** | - | 0.0 | 0.40 | 0.85 | 1.0 | **0.96** | - | 0.71 | 0.70 | 0.62 | 0.62 |
| | 50k | **0.98** | - | 0.28 | 0.72 | 0.66 | 0.32 | **0.10** | - | 0.0 | 0.55 | 0.70 | 0.50 | **0.98** | - | 0.71 | 0.70 | 0.64 | 0.53 |
| Case 3 | 5k | **0.97** | - | 0.16 | 0.50 | 0.54 | 0.47 | **0.05** | - | 0.70 | 0.23 | 0.28 | 0.25 | **0.94** | - | 0.44 | 0.28 | 0.71 | 0.75 |
| | 10k | **0.99** | - | 0.15 | 0.53 | 0.57 | 0.47 | **0.08** | - | 0.75 | 0.28 | 0.28 | 0.25 | **0.94** | - | 0.40 | 0.31 | 0.71 | 0.75 |
| | 50k | **1.0** | - | 0.15 | 0.70 | 0.52 | 0.45 | **0.0** | - | 0.75 | 0.18 | 0.30 | 0.25 | **1.0** | - | 0.40 | 0.28 | 0.67 | 0.75 |
| Case 4 | 5k | **0.93** | - | 0.02 | 0.21 | 0.0 | 0.23 | **0.40** | - | 0.17 | 0.30 | 1.0 | 0.29 | **0.63** | - | 0.03 | 0.33 | 0.0 | 0.54 |
| | 10k | **0.98** | - | 0.02 | 0.24 | 0.0 | 0.23 | **0.29** | - | 0.17 | 0.36 | 1.0 | 0.29 | **0.71** | - | 0.03 | 0.28 | 0.0 | 0.54 |
| | 50k | **0.97** | - | 0.0 | 0.39 | 0.0 | 0.23 | **0.17** | - | 1.0 | 0.21 | 1.0 | 0.29 | **0.82** | - | 0.0 | 0.31 | 0.0 | 0.54 |

*II. Alternatively, if there are no latent confounders, the entire causal graph can be identified with aid of only Proposition 1, a simplified version of Theorem 1 from Phase I.*

## 4 EXPERIMENTS

### 4.1 SYNTHETIC DATA

In the simulation studies, we compared our method with competitive baselines, including typical LiNGAM (Shimizu et al., 2006; 2011), measurement-based methods such as BPC (Silva et al., 2006) and FastGIN (Xie et al., 2020), hierarchical methods such as LaHME (Xie et al., 2022) and IL²H (Huang et al., 2022). We considered four typical causal structures: *i)* all measured variables, *ii)* latent mediators between measured variables, *iii)* latent hierarchies, and *iv)* a general causal structure synthesized from first three cases, shown in Figure 2 . To evaluate the results, we employed three metrics adapted from Xie et al. (2022) and Huang et al. (2022), including *i)* Correct Ordering Rate, *ii)* Error Rate in Latent Variables, and *iii)* F1-score. For more information, please refer to Appendix G. The experimental results are reported in Table 2. Our algorithm performs well with all structures, which proves that it can deal with general causal structures, whereas other methods focus only on particular settings. For complex causal structures such as Case 4, a substantial sample size is often required. One possible reason is that the reliable estimation of higher-order statistics requires much more samples than that of second-order statistics (Hyvärinen et al., 2009).

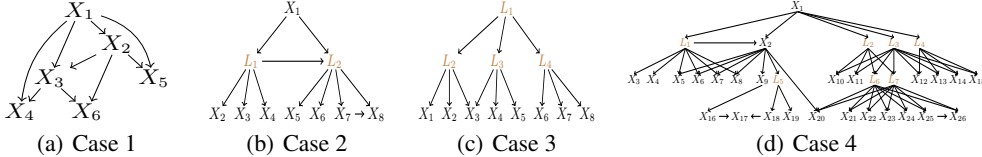

(a) Case 1      (b) Case 2      (c) Case 3      (d) Case 4

Figure 2: Causal structures used in our simulation studies. Case 4 is the same as in Figure 1.

### 4.2 REAL-WORLD DATA

We evaluate our method on two real-world datasets, including H&S-1939 dataset (Holzinger & Swineford, 1939) and SOFA dataset (Muthén & Muthén, 2017). More details is in Appendix H. **Result on H&S-1939 dataset:** We focus on nine out of the original 26 tests, as in Cui et al. (2019). The result graph is shown in Figure 10(a). Consistent with the path diagram in Cui et al. (2019), the variables belonging to the textual aspect ($T_1$, $T_2$, $T_3$) and visual aspect ($V_1$, $V_2$, $V_3$) are clustered and influenced by their respective latent factors. Furthermore, the variables in the speeded aspect ($S_1$, $S_2$, $S_3$) are influenced by those of textual aspect and visual aspect. This relationship can be attributed to the fact that speed is, in essence, a reflection of both textual and visual comprehension. **Result on SOFA dataset:** The result graph of our output forms a hierarchical structure, consistent with the hypothesized model given in Chapter 5 of Muthén & Muthén (2017).

## 5 CONCLUSIONS AND FUTURE WORK

In this paper, we have theoretically demonstrated the identifiability of causal structures under the linear causal model and non-Gaussianity assumptions, without prior knowledge regarding the presence or positions of latent variables. Furthermore, we proposed a feasible iterative algorithm to identify the causal graph. To strike a good balance between theoretical identifiability and practical feasibility we currently require a sufficient number of pure children for each atomic unit. One future research is directed towards reducing the number of pure children required. Another direction for future research is to estimate the causal structure under nonlinear causal models.

## ACKNOWLEDGMENTS

This material is based upon work supported by the AI Research Institutes Program funded by the National Science Foundation under AI Institute for Societal Decision Making (AI-SDM), Award No. 2229881. The project is also partially supported by the National Institutes of Health (NIH) under Contract R01HL159805, and grants from Apple Inc., KDDI Research Inc., Quris AI, and Infinite Brain Technology. FX would like to acknowledge the support by the Natural Science Foundation of China (62306019).

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

# A  RELATED WORK

Inferring causal relationships from observed data is a popular field of statistics and artificial intelligence. As described in the introduction, from the scope of application, most methods can be divided into three areas: a) Infer the causal relations among measured variables with the assumption that there are no latent variables, for example, PC algorithm (Spirtes & Glymour, 1991), LiNGAM-based method (Shimizu et al., 2006), Greedy search (Chickering, 2002). b) Infer the causal relations among measured variables in the presence of latent variable interference, for example, FCI and its variants (Spirtes et al., 1995; Colombo et al., 2012; Claassen et al., 2013). c) Infer the causal relations among latent variables, for example, measurement models (Silva et al., 2006; Kummerfeld & Ramsey, 2016; Cai et al., 2019; Xie et al., 2020) and hierarchical models (Pearl, 1988; Zhang, 2004; Choi et al., 2011; Drton et al., 2017; Xie et al., 2022; Huang et al., 2022; Kong et al., 2023).

Depending on the different constraints and criteria used, most previous causal discovery methods can be divided into the following main categories:

1.) (Non-parametric setting) Conditional independence constraint: The PC algorithm (Spirtes & Glymour, 1991), as well as the FCI algorithm (Spirtes et al., 1995) and their variants (Colombo et al., 2012; Claassen et al., 2013), are prominent examples of non-parametric methods that infer causal relationships among measured variables by testing conditional independence. They do not care about whether the causal relationships are linear or not. However, these methods struggle with the latent variables and their causal relationships. Another limitation is that they yield multiple Markov equivalent structures, introducing ambiguity regarding the presence of confounders in the resulting graph.

2.) (Linear Gaussian setting) Tetrad condition and rank deficiency constraint: By investigating the sub-covariance matrix of measured variables, Tetrad condition (Spearman, 1928) allows to locate latent variables and infer the causal relationships among them in linear-Gaussian settings (Pearl, 1988; Silva et al., 2006; Kummerfeld & Ramsey, 2016). These methods generally assume that each latent variable possesses at least three pure measured variables as pure children, and each measured variable is exclusively a pure child of a single latent variable. Squires et al. (2022) presented a Tetrad-based method that allows latent variables to be influenced by measured variables. It relies on strong assumptions, such as the absence of edges between pairs of observed nodes or between pairs of latent nodes. The rank deficiency constraint is a general version of the Tetrad condition. Huang et al. (2022) use it to infer a latent hierarchical structure, which assumes k+1 pure children and k+1 neighbors for a set of latent variables with size k, and implicitly assumes all clustered children are pure children. Similar to the conditional independence constraint-based methods, they return a result that is asymptotic to the ground truth graph.

3.) (Linear non-Gaussian setting) Independent component analysis (ICA): Shimizu et al. (2006) leveraged non-Gaussianity of data and showed that a linear non-Gaussian acyclic model (LiNGAM) is identifiable based on ICA algorithm. By over-complete ICA, for example, Hoyer et al. (2008); Shimizu et al. (2009); Tashiro et al. (2014) learn the causal structure with latent variables. The limitation of over-complete ICA is that there are equivalence structures and the estimation is easy to fall into local optima. Recently, based on overcomplete ICA, Adams et al. (2021) established necessary and sufficient conditions for structure identifiability in both the linear non-Gaussian and the linear heterogeneous setting. However, it does not have a practical estimation approach and requires knowing the number of latent variables at the beginning.

4.) (Linear non-Gaussian setting) Independent noise, Triad constraint, and GIN condition: Shimizu et al. (2011) proposed DirectLiNGAM in 2011, which learns LiNGAM by replacing the ICA algorithm with regression and independence tests of noises. Triad constraint (Cai et al., 2019) is an extension of the independent noise condition for latent variables with the same assumption that the noise terms are non-Gaussian, and is used to discover the structure of latent variables. GIN condition Xie et al. (2020) is a more general version that allows multiple parents behind multiple children. Xie et al. (2022) identify a latent hierarchical structure by using the GIN condition, which assumes one-factor clusters and measured variables to be descendants of latent variables.

5.) (Linear setting) Matrix decomposition and Optimization: Matrix decomposition techniques have been applied to model causal structures. Under specific conditions, the precision matrix can be decomposed into two components: a low-rank matrix that characterizes the causal relationships from

latent variables to measured variables, and a sparse matrix that represents the relationships among measured variables (Chandrasekaran et al., 2010; 2011). By requiring triple measured variables than latent variables, Anandkumar et al. (2013) decompose the covariance matrix, and is capable of handling DAGs with effective depth one and multi-level Directed Acyclic Graphs (DAGs) (similar to latent hierarchical structures), with the help of low-order observable moments.

6.) Others: With the help of discrete latent variables, Kivva et al. (2021) proposed a mixture oracles-based method to identify the potentially nonlinear latent variable graph. Recently, Kong et al. (2023) learns a nonlinear latent hierarchical causal model by self-supervised representation learning. This innovative method accommodates general nonlinearity and multi-dimensional continuous variables. However, this approach still relies on strong assumptions regarding the data generation process and the estimation is easy to fall into local optima.

# B WHEN THE PO-LINGAM IDENTIFIABILITY CONDITION IS VIOLATED

## B.1 THEORETICAL ANALYSIS

**PO-LiNGAM Identifiability Condition**: a.) Any latent variable is in at least one atomic unit. b.) For any two atomic units with overlapping variables, their non-overlapping parts do not influence each other.

(i) Violation of identifiability condition(a): Our algorithm discovers new atomic units by clustering their individual pure children, and whether the number of pure children for latent variables satisfies the condition is tested during the discovery process, which is mainly achieved by Theorem 2, Remark 2, and Corollary 1. Meanwhile, please notice that our algorithm sequentially identifies the whole causal graph from leaves to roots. The higher-order causal structure can be inferred only after the lower-order causal structure are fully identified. As a consequence, the algorithm halts its inference of the higher-order causal structure when the latent set lacks a sufficient number of individual pure children to be located, so that it does not output spurious latent variables. In other word, since this condition is violated, it can not detect all latent variables, but we can still trust the discovered ones. The atomic units, whose causal orders are lower or equal to the unidentified latent set, and the causal relations among them are still identifiable.

(ii) Violation of identifiability condition(b): Let's consider the example in Figure 3(a), where two atomic units $\{L_1, L_2\}$ and $\{L_2, L_3\}$ are both caused by $\{X_1\}$. Meanwhile, $L_1$ causes $L_3$, which violates identifiability condition(b). In our algorithm, for any two atomic units with overlapping variables, it ignores the influence of non-overlapping part of one atomic unit to that of another and infer their causal relationships separately, as illustrated in Remark 1 and Remark 2. Therefore, the discovered causal graph will ignore the causal relationships between non-overlapping parts of any two overlapping atomic units, as shown in Figure 3(b).

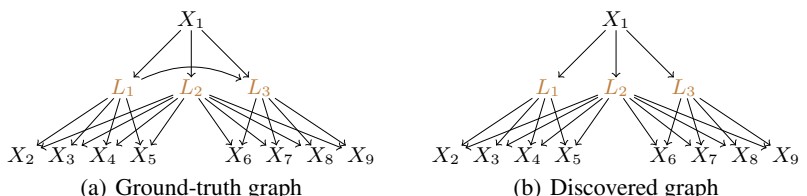

| (a) Ground-truth graph | (b) Discovered graph |

Figure 3: An example of causal graphs where identifiability condition(b) is violated.

## B.2 EXPERIMENTS

(i) To evaluate the performance of our algorithm under the violation of condition(a), we adjusted the causal graphs in Cases 2 and 3 of the simulation experiment, resulting in Case A and Case B of Figure 4. In Case A, $L_1$ lacks sufficient pure atomic unit children for discovery due to the causal chain $X_2 \to X_3 \to X_4$. Nevertheless, the algorithm can still identify the causal structure of $L_1$ and its descendants, as shown in Figure 4(b). Moreover, in Case B, where $L_2$ cannot be discovered due to having only one pure atomic unit child, namely $X_1$, the algorithm still identifies the causal structure of $L_3$ and $L_4$, as shown in Figure 4(d). The corresponding average numerical result from

ten executions is presented in Table 3. The high Correct Ordering Rate indicates that the discovered partial causal graph is highly accurate.

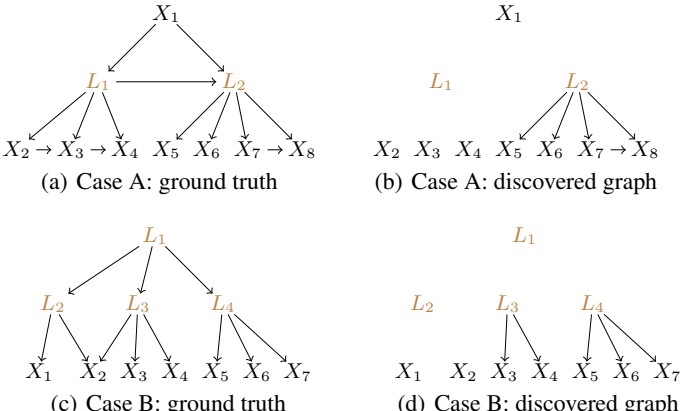

(a) Case A: ground truth          (b) Case A: discovered graph

(c) Case B: ground truth          (d) Case B: discovered graph

Figure 4: Causal graphs that violate identifiability condition(a), and the corresponding discovered results acquired by our method.

Table 3: The performance of our method on three causal graphs that violate the PO-LiNGAM identifiability condition. Case A and Case B violate the condition(a), as shown in Figure 4. Case C violates the condition(b), as shown in Figure 3.

|  | Case A | | | Case B | | | Case C | | |
|---|---|---|---|---|---|---|---|---|---|
|  | 5k | 10k | 50k | 5k | 10k | 50k | 5k | 10k | 50k |
| Correct Ordering Rate ↑ | **0.98** | **1.0** | **1.0** | **1.0** | **1.0** | **1.0** | **0.94** | **0.94** | **1.0** |
| Error Rate in Latent Variables ↓ | 0.45 | 0.5 | 0.5 | 0.53 | 0.5 | 0.5 | 0.66 | 0.52 | 0.33 |
| F1-score ↑ | 0.55 | 0.56 | 0.56 | 0.60 | 0.62 | 0.62 | 0.54 | 0.66 | 0.85 |

(ii) To evaluate the performance of our algorithm under the violation of condition(b), we tested the algorithm with the ground-truth graph in Figure 3(a). The result discovered by our method is in Figure 3(b). The corresponding average numerical result from ten executions is presented in Case C of Table 3. The high Correct Ordering Rate indicates that the discovered partial causal graph is highly accurate.

## C  DETAILS OF IDENTIFICATION ALGORITHM

Our algorithm is iterative, with each iteration comprising three main phases, progressively uncovering the entire causal graph from leaf to root nodes. For each iteration, it first recursively identifies the leaf nodes and their parent nodes from the active atomic unit set $\mathbb{A}$ in the causal graph composed of atomic units of $\mathbb{A}$ and their latent confounders, and update $\mathbb{A}$ (**Phase I**). Then, we discover new atomic units by clustering their pure children in $\mathbb{A}$, and update $\mathbb{A}$ (**Phase II**). After that, we refine newly discovered atomic units by checking overlapping atomic units and decomposable 'atomic units', and update $\mathbb{A}$ (**Phase III**). At the end of an iteration, $\mathbb{A}$ is sent back to the Phase I and a new iteration starts until no more atomic units in $\mathbb{A}$ are updated.

### C.1  PHASE I

The detailed algorithm for Phase I is in Algorithm 2.

### C.2  PHASE II

The detailed algorithm for Phase II is in Algorithm 3.

### C.3  PHASE III

The detailed algorithm for Phase III is in Algorithm 4.

### C.4 Supplementary algorithms for phase I

To handle the situation explained at the end of section 3.2, we proposed two additional steps for Phase I, including Algorithm 5 (line 8, Alg. 2) and Algorithm 6 (line 14, Alg. 2).

Small atomic units that can be completely covered by larger atomic units are sometimes indistinguishable from those larger atomic units due to the different causal order counting from the leaf nodes of the graph as well as the characterization of the GIN condition. The purpose of Algorithm 5 is to check this case by rechecking the number of parent variables of the related atomic unit children. If such a case happens, most children are re-added into $\mathbb{A}$ and treated the same as others in $\mathbb{A}$. For the special children that have partial parents not in $\mathbb{A}$, Algorithm 6 is used to relocate parents for them after the normal children are settled.

### C.5 Check for multiple root atomic units

Most causal discovery methods for handling latent variables Silva et al. (2006); Cai et al. (2019); Xie et al. (2020; 2022); Huang et al. (2022) implicitly assume that there are no unconditional independence between any two variables in the measured variable set $\mathbf{X}$. In other words, they assume that there is at most one root node in the causal graph, which may be a latent variable or a latent variable set with common children.

In practical situations, there may be more than one root node in a causal graph, not to mention that the input measured variables may form multiple disconnected subgraphs. In Figure 5(a), $X_1 \ldots X_4$ form a subgraph, while $X_5$ is not connected to any other variables. Furthermore, in the sub causal graph consisting of $X_1 \ldots X_4$, $X_1$ is unconditionally independent of both $X_3$ and $X_4$. In such a case, there are three root nodes: $X_1$, $L_1$, and $X_5$.

Our methods can handle the situation regardless of the number of root atomic units in the causal graph. At the end of the inference process, the algorithm tends to cluster all the true root atomic units (three or more) together and assigns them a common latent parent variable, as depicted in Figure 5(b). This occurs because when the active atomic unit set $\mathbb{A}$ contains only root atomic units, those root atomic units naturally satisfy the GIN condition and are unconditionally independent of each other without requiring any additional operations. It is considered that those root atomic units share a common latent parent variable with a value of 0. To verify the presence of multiple root atomic units in the causal graph, we can perform the independence tests. If the discovered root atomic unit comprises only one latent variable and lacks impure children, we should examine the unconditional independence of the pure atomic unit children of that discovered root atomic unit (For latent atomic units, their measured surrogate sets are used). If these pure children are found to be unconditionally independent of each other, then we can conclude that they are the true root atomic units of the causal graph. One the other hand, if the initially discovered causal graph has only two root atomic units, we can do the same unconditional independence test to check whether there is a common latent variable parent behind them.

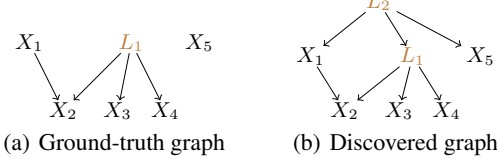

(a) Ground-truth graph     (b) Discovered graph

Figure 5: An example of causal structures where there are multiple root atomic units.

---

**Algorithm 2:** Phase I: IdentifyLeafNodesAndTheirParents

---

**Input:** Partial causal graph $\mathcal{G}$, Active atomic unit set $\mathbb{A}$
**Output:** Partial causal graph $\mathcal{G}$, Active atomic unit set $\mathbb{A}$

1 **repeat**
2      Select an atomic unit $\mathcal{U} \in \mathbb{A}$;
3      **for** *GrLen $\leftarrow$ 1 to $|\mathbb{A}\backslash\mathcal{U}|$* **do**
4          **repeat**
5              Select atomic unit subset $\mathbb{P}$ from $\mathbb{A}\backslash\mathcal{U}$ such that $|\mathbb{P}| = GrLen$;
6              **if** $\mathcal{U}$ *with* $\mathbb{P}$ *satisfies Theorem 1 and Remark 1* **then**
7                  $\mathbb{A} \leftarrow \mathbb{A}\backslash\mathcal{U}$, and update $\mathcal{G}$;
8                  $(\mathcal{G}, \mathbb{A}) \leftarrow \text{CheckSubAtomicUnits}(\mathcal{G}, \mathbb{A}, \mathbb{P})$;
9                  Return to line 2;
10              **end**
11          **until** *all subsets with size GrLen in $\mathbb{A}\backslash\mathcal{U}$ selected*;
12      **end**
13 **until** *no more atomic units can be removed from $\mathbb{A}$*;
14 $(\mathcal{G}, \mathbb{A}) \leftarrow \text{RefindChildrenForSubAtomicUnits}(\mathcal{G}, \mathbb{A})$;
15 **return** $\mathcal{G}, \mathbb{A}$

---

**Algorithm 3:** Phase II: DiscoverNewAtomicUnits

---

**Input:** Partial causal graph $\mathcal{G}$, Active atomic unit set $\mathbb{A}$
**Output:** Partial causal graph $\mathcal{G}$, Active atomic unit set $\mathbb{A}$

1 GrLen:= 2, $\mathbb{A}' := \mathbb{A}$, and NewAtomicUnitSet := $\emptyset$;
2 **repeat**
3      ClusterList:= $\emptyset$;
4      **repeat**
5          Select subset $\mathbf{Y}$ from $\text{MS}_b(\mathbb{A}')$ such that $\|\mathbf{Y}\| = \text{GrLen}$, $\mathbf{Y} \subseteq \text{MS}_b(\mathbb{Y})$ and $\mathbb{Y} \geq 2$;
6          **if** $\mathbf{Y}$ *satisfy the Theorem 2 and Remark 2* **then**
7              Add $\mathbb{Y}$ into ClusterList;
8          **end**
9      **until** *all subset with size GrLen in $\text{MS}_b(\mathbb{A}')$ selected*;
10      Gather the sets having overlapping atomic units with each other from the ClusterList as a set of sets and add it into another set $\mathbf{C}$, repeatedly and exhaustively;
11      **for** *each $\mathbf{C}_i \in \mathbf{C}$* **do**
12          **when** GrLen= 2, **if** $\mathbf{C}_i$ contains only one set (bi-unit cluster) **then** identify pure children by Lemma 1 in Xie et al. (2022);
13          Select one set $\in \mathbf{C}_i$, remove one atomic unit from it, get a base $\mathbb{B}$ with $\|\mathbb{B}\| \geq \text{Grlen-1}$;
14          PureChildSet = $\emptyset$; **for** each set $\mathbb{S} \in \mathbf{C}_i$ **if** $|\mathbb{S}| = |\mathbb{B}| + 1$ **do** add $\mathbb{S}\backslash\mathbb{B}$ into PureChildSet;
15          **for** *each atomic unit $\mathcal{B} \in \mathbb{B}$* **do**
16              Select Grlen-1 atomic units from PureChildSet and patch $\mathcal{B}$ with them to form set $\mathbb{T}$;
17          **end**
18          **if** *bi-unit cluster is pure cluster, or $|PureChild| \geq Grlen-1$ and all $\mathbb{T}s \in \mathbf{C}_i$* **then**
19              The atomic units in PureChildSet and $\mathbb{B}$ are all pure children;
20              $\mathbb{A} \leftarrow \mathbb{A}\backslash\mathbb{T}\backslash\text{PureChildSet}$, and update $\mathcal{G}$;
21              Create a new atomic unit $\mathcal{U}$ and save it in NewAtomicUnitSet;
22          **end**
23      **end**
24      $\mathbb{A}' \leftarrow \mathbb{A}'\backslash\text{ClusterList}$, and GrLen $\leftarrow$ GrLen+1;
25 **until** *no more clusters are found*;
26 **return** $\mathcal{G}, \mathbb{A}$, and NewAtomicUnitSet

---

---

**Algorithm 4:** Phase III: RefineAtomicUnits

**Input:** Partial causal graph $\mathcal{G}$, Active atomic unit set $\mathbb{A}$, NewAtomicUnitSet
**Output:** Partial causal graph $\mathcal{G}$, Active atomic unit set $\mathbb{A}$

1 **repeat**
2    Select two atomic unit $\mathcal{U}_1, \mathcal{U}_2 \in \mathbb{A}$, where at least one in NewAtomicUnitSet;
3    **for** $m \leftarrow 1$ **to** $\|\mathcal{U}_1\|$ **do**
4      Select m measured surrogates from $\mathrm{MS}_b(\mathcal{U}_1)$, denoted as $\mathbf{U}_1$;
5      **if** $(\mathrm{MS}_a(\mathcal{U}_1 \cup \mathcal{U}_2),\ \mathbf{U}_1 \cup \mathrm{MS}_b(\mathcal{U}_2))$ *follows GIN condition* **then**
6        $\mathcal{U}_1$ and $\mathcal{U}_2$ have $\|\mathcal{U}_1\| - \|\mathbf{U}_1\| + 1$ overlapping variables;
7        Update $\mathcal{G}$, and merge $\mathcal{U}_1$ with $\mathcal{U}_2$;
8        Return to line 2;
9      **end**
10    **end**
11 **until** *all binary atomic unit set selected*;
12 **for** *each atomic unit* $\mathcal{U} \in \mathbb{A}$ **do**
13    **for** each $\mathcal{S}_i \in \mathbb{A} \backslash \mathcal{U}$ and $\mathcal{S}_i \subseteq \mathcal{U}$ **do** add $\mathcal{S}_i$ into set $\mathbb{C}$;
14    **repeat**
15      Select a subset $\mathbb{S} \subseteq \mathbb{C}$;
16      **if** *atomic units in* $\mathbb{S}$ *do not overlap and* $\|\mathcal{U}\| = \|\mathbb{S}\|$ **then**
17        Move the children of $\mathcal{U}$ to below the atomic units in $\mathbb{S}$;
18        **if** $\mathbb{A}$ is not updated, **then** $\mathbb{A} \leftarrow \mathbb{A} \backslash \mathcal{U}$;
19        Update $\mathcal{G}$;
20      **end**
21    **until** *all subset in* $\mathbb{C}$ *selected*;
22 **end**
23 **return** $\mathcal{G}, \mathbb{A}$

---

**Algorithm 5:** CheckSubAtomicUnits

**Input:** Partial causal graph $\mathcal{G}$, Active atomic unit set $\mathbb{A}$, atomic unit set $\mathbb{P} \subseteq \mathbb{A}$
**Output:** Partial causal graph $\mathcal{G}$, Active atomic unit set $\mathbb{A}$

1 Let $\widehat{\mathbb{P}}$ be the union of $\mathbb{P}$ and the atomic units in $\mathbb{A}$ that overlap with atomic units in $\mathbb{P}$;
2 Let $\mathbb{U}$ contains the atomic units whose parents is a subset of $\widehat{\mathbb{P}}$;
3 GrLen:= 2, $\mathbb{U}' := \mathbb{U}$;
4 **repeat**
5    ClusterList:= $\emptyset$;
6    **repeat**
7      Select subset $\mathbf{Y}$ from $\mathrm{MS}_b(\mathbb{U}')$ such that $\|\mathbf{Y}\| = $ GrLen, $\mathbf{Y} \subseteq \mathrm{MS}_b(\mathbb{Y})$ and $\mathbb{Y} \geq 2$;
8      **if** $\mathbf{Y}$ *satisfy the Theorem 2 and Remark 2* **then**
9        Add $\mathbb{Y}$ into ClusterList;
10        **if** $\|\mathbf{Y}\| \leq$ *known* $\|\mathrm{Pa}(\mathbb{Y})\|$ **then**
11          **if** there are atomic units $\notin \mathbb{A}$, which has partial atomic unit parents in $\widehat{\mathbb{P}}$ and the rest atomic unit parents $\notin \mathbb{A}$, **then** mark them and the atomic units in $\mathbb{U}$ with matching symbols;
12          $\mathbb{A} \leftarrow (\mathbb{A} \backslash \widehat{\mathbb{P}}) \cup \mathbb{U}$, update $\mathcal{G}$, and go to line 18;
13        **end**
14      **end**
15    **until** *all subset with size GrLen in* $\mathrm{MS}_b(\mathbb{U}')$ *selected*;
16    $\mathbb{U}' \leftarrow \mathbb{U}' \backslash$ ClusterList, and GrLen $\leftarrow$ GrLen+1;
17 **until** *no more clusters are found*;
18 **return** $\mathcal{G}, \mathbb{A}$

---

---

**Algorithm 6:** RefineChildrenForSubAtomicUnits

---

**Input:** Partial causal graph $\mathcal{G}$, Active atomic unit set $\mathbb{A}$
**Output:** Partial causal graph $\mathcal{G}$, Active atomic unit set $\mathbb{A}$

**1 for** *each marked set of atomic units in $\mathcal{G}$* **do**

**2**     **if** *the atomic units in $\mathbb{U}$ are all relocated to be children of the smaller atomic units* **then**

**3**        Relocate the parents for the atomic units that used to have partial atomic unit parents in $\widehat{\mathbb{P}}$ and the rest of atomic unit parents $\notin \mathbb{A}$, by Theorem 1 and Remark 1;

**4**        Update $\mathcal{G}$;

**5**     **end**

**6 end**

**7 return** $\mathcal{G}, \mathbb{A}$

---

# D ILLUSTRATION OF OUR ALGORITHM

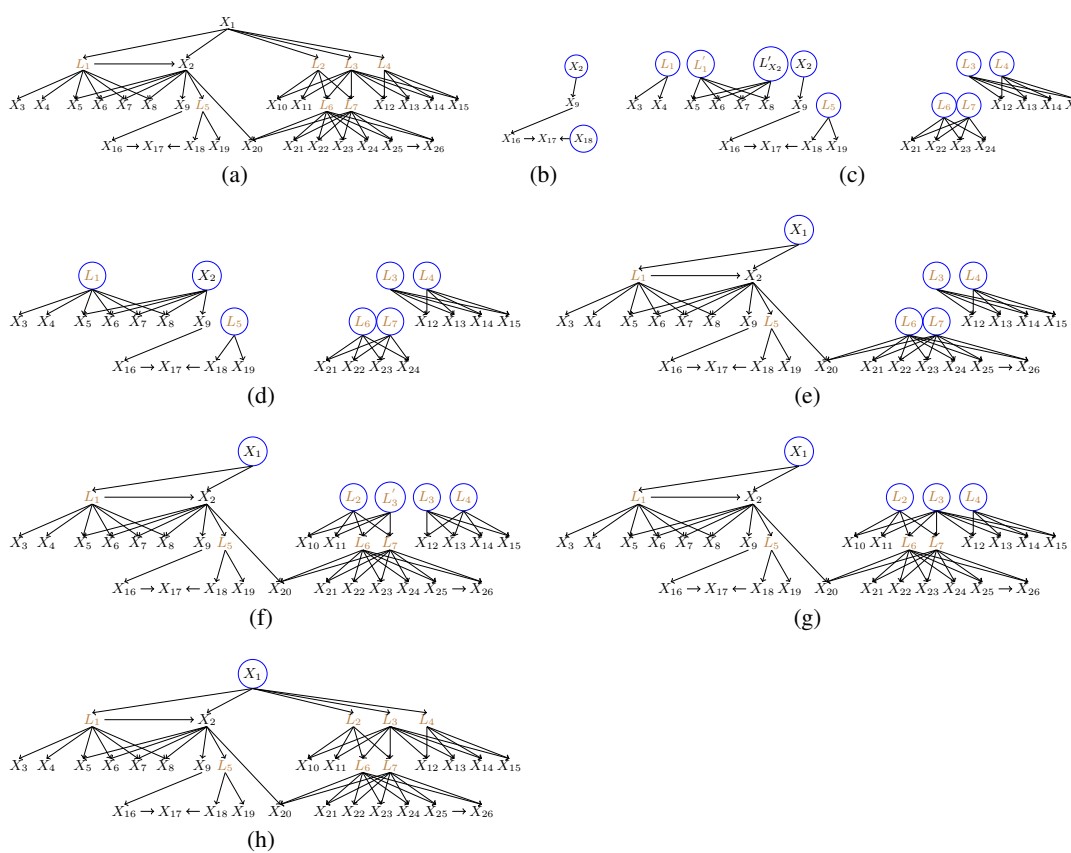

Figure 6: Illustration of the entire procedure of our algorithm. For each subgraph, the nodes in blue circles are active variables that are under investigation, and other measured variables for which causal relationships have not been identified are not shown for clarity. (a) Ground-truth structure. (b) Output after phase I of the first iteration of our algorithm. (c) Output after phase II of the first iteration. (d) Output after phase III of the first iteration. (e) Output after phase I of the second iteration. (f) Output after phase II of the second iteration. (g) Output after phase III of the second iteration. (h) Output after phase I of the third iteration, which has recovered the true causal structure.

In this section, we illustrate our algorithm with the ground-truth graph in Figure 6(a). We assume Oracle tests for GIN conditions. In this structure, $L_1, \ldots, L_7$ are latent variables and $X_1, \ldots, X_{26}$ are measured variables. We can see from the graph that measured variables are interspersed with

latent variables. The causal relationships among them are also complex. The estimation process is as follows:

i Initially, the active atomic unit set $\mathbb{A} = \{\{X_1\}, \ldots, \{X_{26}\}\}$ and graph $\mathcal{G} = \emptyset$.

ii During phase I of the first iteration, it first identifies $\mathrm{Pa}(\{X_{17}\}) = \{\{X_{16}\}, \{X_{18}\}\}$, and remove $\{X_{17}\}$ from the active set $\mathbb{A}$. After that, we sequentially identify $\mathrm{Pa}(\{X_{16}\}) = \{X_9\}$ and $\mathrm{Pa}(\{X_9\}) = \{X_2\}$, and update $\mathbb{A}$. The result after phase I of the first iteration is shown in Figure 6(b), and the current $\mathbb{A} = \{\{X_1\}, \ldots, \{X_8\}, \{X_{10}\}, \ldots, \{X_{15}\}, \{X_{18}\}, \ldots, \{X_{26}\}\}$.

iii During phase II of the first iteration, we cluster the individual pure children in $\mathbb{A}$ to form new atomic units. It is worth noting that we do not know the true serial numbers of the latent variables. In the following, for the sake of clarity, we use the true serial numbers of latent variables to denote them. As shown in Figure 6(c), we introduce the new atomic unit $\{L_1\}$ to be parent of individual pure children set $\{\{X_3\}, \{X_4\}\}$, the new atomic unit $\{L'_1, L'_{X_2}\}$ to be parent of individual pure children set $\{\{X_5\}, \{X_6\}, \{X_7\}, \{X_8\}\}$, the atomic unit $\{L_3, L_4\}$ to be parent of individual pure children set $\{\{X_{12}\}, \{X_{13}\}, \{X_{14}\}, \{X_{15}\}\}$, the atomic unit $\{L_5\}$ to be parent of individual pure children set $\{\{X_{18}\}, \{X_{19}\}\}$, the atomic unit $\{L_6, L_7\}$ to be parent of individual pure children set $\{\{X_{21}\}, \{X_{22}\}, \{X_{23}\}, \{X_{24}\}\}$. We add the newly discovered atomic units into $\mathbb{A}$ and remove their individual pure children from $\mathbb{A}$.

iv During phase III of the first iteration, we find that the newly discovered atomic unit $\{L'_1, L'_{X_2}\}$ can be fully replaced by $\{L_1\}$ and $\{X_2\}$, as shown in Figure 6(d). We move the children of $\{L'_1, L'_{X_2}\}$ to $\{L_1\}$ and $\{X_2\}$, and remove $\{L'_1, L'_{X_2}\}$ from $\mathbb{A}$.

v Now, we start the second iteration. During phase I of the second iteration, we find $\mathrm{Pa}(\{L_5\}) = \{X_2\}$, $\mathrm{Pa}(\{X_{20}\}) = \{\{X_2\}, \{L_6, L_7\}\}$, $\mathrm{Pa}(\{X_2\}) = \{\{L_1\}, \{X_1\}\}$, $\mathrm{Pa}(\{L_1\}) = \{X_1\}$, $\mathrm{Pa}(\{X_{26}\}) = \{\{X_{25}\}, \{L_6, L_7\}\}$, and $\mathrm{Pa}(\{X_{25}\}) = \{L_6, L_7\}$, as shown in Figure 6(e).

vi In phase II of the second iteration, we introduce the new atomic unit $\{L_2, L'_3\}$ to be parent of individual pure children set $\{\{X_{10}\}, \{X_{11}\}, \{L_6, L_7\}\}$, as shown in Figure 6(f).

vii In phase III of the second iteration, we find that the newly discovered atomic unit $\{L_2, L'_3\}$ have one variable overlapping with atomic unit $\{L_3, L_4\}$, and merge them. The result is shown in Figure 6(g).

viii Now, we start the third iteration. During phase I of the third iteration, we find $\mathrm{Pa}(\{L_2, L_3\}) = \{X_1\}$ and $\mathrm{Pa}(\{L_3, L_4\}) = \{X_1\}$, as shown in Figure 6(h). After that, the current active atomic unit set contains only the measured variable $\{X_1\}$, and the whole underlying causal graph is identified.

## E  COMPUTATIONAL COMPLEXITY OF OUR ALGORITHM

In this section, we analyze the complexity of our algorithm. Denote by $n$ the number of active atomic units at the beginning of each phase. The complexity of phase I is upper bounded by $\mathcal{O}(n \sum_{k=1}^{n-1} \binom{n-1}{k})$. The phase II has the worst case complexity $\mathcal{O}(\sum_{k=2}^{n} \binom{n}{k})$. Denote by $l$ the size of the largest atomic unit in the active atomic unit set. The complexity of phase III is upper bounded by $\mathcal{O}(l\binom{n}{2})$. The total complexity of the algorithm to discover the whole causal graph depends on the number of (both measured and latent) variables and the structural density of the causal graph, which determine how many iterations the algorithm needs to run.

## F  PROOFS

Before presenting the proofs of our results, we need a few more theorems and definitions derived from Xie et al. (2020; 2023).

**Definition 6** (GIN condition (Xie et al., 2020)). *Let $\mathbf{Z}$, $\mathbf{Y}$ be sets of variables in a linear non-Gaussian acyclic causal model. We say that $(\mathbf{Z}, \mathbf{Y})$ follows GIN condition if and only if $\omega^\intercal \mathbf{Y}$ are statistically independent of $\mathbf{Z}$, where $\omega$ satisfies $\omega^\intercal \mathbb{E}[\mathbf{Y}\mathbf{Z}^\intercal] = 0$ and $\omega \neq 0$.*

In other words, $(\mathbf{Z}, \mathbf{Y})$ violates the GIN condition if and only if $E_{\mathbf{Y}||\mathbf{Z}}$ is dependent on $\mathbf{Z}$.

Recently, Xie et al. (2023) extended the original graphical criteria of the GIN condition with the help of trek and trek-separation (t-separation) (Sullivant et al., 2010). We next describe the notion of the trek and trek-separation criterion, which is more general than d-separation in linear causal models. After that, we show the graphical implication of the GIN condition in PO-LiNGAM, which helps to exploit the GIN condition to discover the causal graph.

**Definition 7** (trek (Sullivant et al., 2010)). *A trek in $\mathcal{G}$ from $\mathbf{i}$ to $\mathbf{j}$ is an ordered pair of directed paths $(\mathbf{P_1}, \mathbf{P_2})$ where $\mathbf{P_1}$ has sink $\mathbf{i}$, $\mathbf{P_2}$ has sink $\mathbf{j}$, and both $\mathbf{P_1}$ and $\mathbf{P_2}$ have the same source $\mathbf{k}$. The common source $\mathbf{k}$ is called the top of the trek, denoted $top(\mathbf{P_1}, \mathbf{P_2})$. Note that one or both of $\mathbf{P_1}$ and $\mathbf{P_2}$ may consist of a single vertex, that is, a path with no edges.*

**Definition 8** (t-separation (Sullivant et al., 2010)). *Let $\mathbf{A}$, $\mathbf{B}$, $\mathbf{C_A}$, and $\mathbf{C_B}$ be four subsets of $\mathbf{V}$. We say the ordered pair ($\mathbf{C_A}$, $\mathbf{C_B}$) t-separates $\mathbf{A}$ from $\mathbf{B}$ if, for every trek ($\tau_1$; $\tau_2$) from a vertex in $\mathbf{A}$ to a vertex in $\mathbf{B}$, either $\tau_1$ contains a vertex in $\mathbf{C_A}$ or $\tau_2$ contains a vertex in $\mathbf{C_B}$.*

**Theorem 5** (GIN Graphical Criteria in PO-LiNGAM). *Let $\mathbf{Y}$ and $\mathbf{Z}$ be two sets of measured variables of a partially observed linear non-Gaussian acyclic causal model (PO-LiNGAM). Assume the rank-faithfulness holds. $(\mathbf{Z}, \mathbf{Y})$ satisfies the GIN condition if and only if there exists a variable set $\mathcal{S}$ with $0 \leq Dim(\mathcal{S}) \leq min(Dim(\mathbf{Y}) - 1, Dim(\mathbf{Z}))$, such that 1) the order pair $(\emptyset, \mathcal{S})$ t-separates $\mathbf{Z}$ and $\mathbf{Y}$, and that 2) the covariance matrix of $\mathcal{S}$ and $\mathbf{Z}$ has rank $Dim(\mathcal{S})$, and so does that of $\mathcal{S}$ and $\mathbf{Y}$.*

**Note**: In this paper, $Dim(\mathcal{S}) = \|\mathcal{S}\|$ represents the number of variables in $\mathcal{S}$, regardless of whether $\mathcal{S}$ is a set of variables or a set of atomic units.

Roughly speaking, the conditions in this theorem can be interpreted in the following way: i.) a causally earlier subset (according to the causal order) of $\mathbf{Y}$ t-separates $\mathbf{Y}$ from $\mathbf{Z}$, and ii.) the linear transformation from that subset of the common causes to $\mathbf{Z}$ has full column rank. Based on the definition of d-separation, the first interpretation is that the causally earlier subset $\mathcal{S}$ (according to the causal order) of $\mathbf{Y}$ d-separates $\mathbf{Y} \backslash \mathcal{S}$ from $\mathbf{Z} \backslash \mathcal{S}$.

*Proof.* Recently, it has been shown in Xie et al. (2023) that the graphical criteria hold in a linear non-Gaussian acyclic causal model. In (Xie et al., 2020; 2022; 2023), they allowed part variables to be latent. In our paper, the latent variable can be the descendent or ancestor of measured variables, which does not affect the graphical criteria of GIN because linear causal models are transitive. □

### F.1 PROOF OF PROPOSITION 1

*Proof.* Let $\mathbf{V}$ be a set of variables satisfying PO-LiNGAM with no latent variables, and V be a variable in $\mathbf{V}$.

(i) Assume that V is a leaf variable of the causal graph, and $\mathbf{P}$ are its all parents. We know that a) $0 \leq \|\mathbf{P}\| \leq \min(\|V \cup \mathbf{P}\| - 1, \|\mathbf{V} \backslash V\|)$, and b) $\mathbf{P}$ is the minimal set so that $(\emptyset, \mathbf{P})$ t-separates $\mathbf{V} \backslash V$ and $V \cup \mathbf{P}$. Furthermore, the set of common components between $V \cup \mathbf{P}$ and $\mathbf{V} \backslash V$ is $\mathbf{P}$, so the covariance matrix of $\mathbf{P}$ and $\mathbf{V} \backslash V$ has rank $\|\mathcal{S}\|$, and so does that of $\mathbf{P}$ and $V \cup \mathbf{P}$. Therefore, $(\mathbf{V} \backslash V, V \cup \mathbf{P})$ follows the GIN condition and there is no $\tilde{\mathbf{P}} \subset \mathbf{P}$ such that $(\mathbf{V} \backslash V, V \cup \tilde{\mathbf{P}})$ follows the GIN condition.

(ii) Assume that $(\mathbf{V} \backslash V, V \cup \mathbf{P})$ follows the GIN condition and there is no $\tilde{\mathbf{P}} \subset \mathbf{P}$ such that $(\mathbf{V} \backslash V, V \cup \tilde{\mathbf{P}})$ follows the GIN condition. We know that $\mathbf{P}$ is the minimal set that $(\emptyset, \mathbf{P})$ t-separates $\mathbf{V} \backslash V$ and $V \cup \mathbf{P}$, and therefore, $\mathbf{P}$ is causal earlier than V. In a causal graph, the minimal causal earlier set that t-separates one variable from the other variables is the parent set of that variable.

Therefore, from (i) and (ii), the proposition is proved. □

### F.2 PROOF OF THEOREM 1

*Proof.* The proof of theorem 1 is similar to the proof of Proposition 1. Let $\mathbb{U}$ be a set of non-overlapping atomic units satisfying PO-LiNGAM with known $\text{MS}_{a,b}$ for each atomic unit in $\mathbb{U}$, and $\mathcal{U}$ be an atomic unit in $\mathbb{U}$.

(i) Assume that $\mathcal{U}$ is a leaf atomic unit in the current causal graph, and $\mathbb{P}$ is the set of parent atomic units of $\mathcal{U}$. We know that a) $0 \leq \|\mathbb{P}\| \leq \min(\|\mathcal{U} \cup \mathbb{P}\| - 1, \|\mathbb{U}\backslash\mathcal{U}\|)$, and b) $\mathbb{P}$ is the minimal set so that $(\emptyset, \mathbb{P})$ t-separates $\mathbb{U}\backslash\mathcal{U}$ and $\mathcal{U} \cup \mathbb{P}$. Furthermore, the set of common components between $\mathcal{U} \cup \mathbb{P}$ and $\mathbb{U}\backslash\mathcal{U}$ is $\mathbb{P}$, so the covariance matrix of $\mathbb{P}$ and $\mathcal{U} \cup \mathbb{P}$ has rank $\|\mathbb{P}\|$, and so does that of $\mathbb{P}$ and $\mathbb{U}\backslash\mathcal{U}$. And because some of the atomic units may be latent, we use their measured surrogate variable set to represent them. Therefore, $\big(\mathrm{MS}_a(\mathbb{U}\backslash\mathcal{U}), \mathrm{MS}_b(\mathcal{U} \cup \mathbb{P})\big)$ follows the GIN condition and there is no $\tilde{\mathbb{P}} \subset \mathbb{P}$ such that $\big(\mathrm{MS}_a(\mathbb{U}\backslash\mathcal{U}), \mathrm{MS}_b(\mathcal{U} \cup \tilde{\mathbb{P}})\big)$ follows the the GIN condition. The same is true if we represent them using another set of measured surrogate variables.

(ii) Assume that $\big(\mathrm{MS}_a(\mathbb{U}\backslash\mathcal{U}), \mathrm{MS}_b(\mathcal{U} \cup \mathbb{P})\big)$ follows the GIN condition and there is no $\tilde{\mathbb{P}} \subset \mathbb{P}$ such that $\big(\mathrm{MS}_a(\mathbb{U}\backslash\mathcal{U}), \mathrm{MS}_b(\mathcal{U} \cup \tilde{\mathbb{P}})\big)$ follows the the GIN condition. $\mathrm{MS}_a(\mathbb{U}\backslash\mathcal{U})$ contains a set of measured surrogate set of $\mathbb{U}\backslash\mathcal{U}$, $\mathrm{MS}_b(\mathcal{U}\cup\mathbb{P})$ contains another set of measured surrogate set of $\mathcal{U}\cup\mathbb{P}$. The common component of $\mathbb{U}\backslash\mathcal{U}$ and $\mathcal{U} \cup \mathbb{P}$ is $\mathbb{P}$. And, without $\mathrm{MS}_b(\mathbb{P})$, the GIN condition cannot be satisfied singly using $\mathrm{MS}_b(\mathcal{U})$ as the $\mathbf{Y}$ set. Thus, $\mathbb{P}$ is the minimal set that $(\emptyset, \mathbb{P})$ t-separates $\mathrm{MS}_a(\mathbb{U}\backslash\mathcal{U})$ and $\mathrm{MS}_b(\mathcal{U} \cup \mathbb{P})$, and therefore, $\mathbb{P}$ is causal earlier than $\mathcal{U}$. In the causal graph, the minimal causal earlier atomic unit set that t-separates one atomic unit from the other atomic units is the parent set of that atomic unit.

Therefore, from (i) and (ii), the theorem is proved. $\square$

### F.3   PROOF OF REMARK 1

*Proof.* (i) The first term is to eliminate the effect of overlapping atomic unit sets when searching for parents, since for any two atomic units with overlapping variables, we don't have enough information to know whether the non-overlapping part of one atomic unit is the parent of the non-overlapping part of another atomic unit. As illustrated in case 1 of Figure 7, If we want to know the parent of the atomic unit $\{L_2, L_3\}$, we need remove the atomic unit $\{L_3, L_4\}$ from $\mathbb{U}$ and $\mathbb{P}$. Otherwise, both $\{L_3, L_4\}$ and $\{L_1\}$ will be considered as the parents of $\{L_2, L_3\}$ rather than the true parent $\{L_1\}$.

(ii) For the second term, let us consider the case 2 of Figure 7, where we want to find the parent of $\{X_9\}$. $\mathbb{P} = \{\{L_1, L_2\}, \{L_2, L_3\}\}$ and $\mathcal{U} = \{X_9\}$. There is no point in testing the GIN condition for $\big(\mathrm{MS}_a(\mathbb{U}\backslash\mathcal{U}), \mathrm{MS}_b(\mathcal{U} \cup \mathbb{P})\big)$, since even without $\mathrm{MS}_b(\mathcal{U})$ in $\mathbf{Y}$ set, the graph criteria of GIN condition is still satisfied, so is GIN condition. That is because those two atomic units $\{L_1, L_2\}$ and $\{L_2, L_3\}$ have overlapping $L_2$, and therefore the unrepeated variable set behind their measured surrogate variable has size 3 instead of 4. Thus, we need to find a subset of $\mathrm{MS}_b(\mathbb{P})$ so that the number of variables is equal to the true size $\|\mathbb{P}\|$. That is the maximal subset $\mathbf{S}$ of $\mathrm{MS}_b(\mathbb{P})$ that makes $\big(\mathrm{MS}_a(\mathbb{U}\backslash\mathcal{U}), \mathbf{S}\big)$ does not follow GIN condition. $\square$

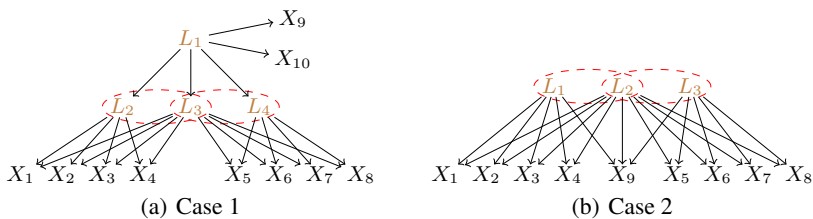

(a) Case 1          (b) Case 2

Figure 7: The illustrative examples for remark 1.

### F.4   PROOF OF THEOREM 2

The aim behind Theorem 2, Remark 2, and Corollary 1 is to find a sufficiently large number of pure children to support the discovery of new atomic units.

*Proof.* Let $\mathbb{U}$ be a set of non-overlapping atomic units satisfying PO-LiNGAM with known $\mathrm{MS}_{a,b}$ for each atomic unit in $\mathbb{U}$, and there is no such leaf atomic unit in $\mathbb{U}$ that its full parents are also in $\mathbb{U}$. Assume that $(\mathrm{MS}_a(\mathbb{U}\backslash\mathbb{Y}), \mathbf{Y})$ follows the GIN condition, and there is no subset $\tilde{\mathbf{Y}} \subset \mathbf{Y}$ such that $(\mathrm{MS}_a(\mathbb{U}\backslash\mathbb{Y}), \tilde{\mathbf{Y}})$ follows the GIN condition. It means that there is a variable set $\mathcal{S}$ with size

$\|\mathbf{Y}\| - 1$, which is causal earlier than $\mathbb{Y}$, and the order pair $(\emptyset, \mathcal{S})$ t-separates $\text{MS}_a(\mathbb{U}\backslash\mathbb{Y})$ and $\mathbf{Y}$. Since we know in $\mathbb{U}$ there is no leaf atomic unit that its full parents are also in $\mathbb{U}$, $\mathcal{S}$ contains at least one latent variable which is the parent of $\mathbf{Y}$. However, $\mathcal{S}$ may also contain variables that are also in $\mathbf{Y}$. The third statement is to check out this situation by explicitly adding the variables in $\mathcal{P}$ into $\mathcal{S}$ and testing the GIN condition. Therefore, with these three statements, we can ensure that behind $\mathbb{Y}$ there is a parent variable set $\mathbf{P}$ involving at least one latent variable, which has size $\|\mathbf{Y}\| - 1$ and can d-separates $\mathbb{Y}$ from the atomic units of $\mathbb{U}\backslash(\mathbb{Y}\cup\mathbf{P})$. $\qquad\square$

### F.5 PROOF OF REMARK 2

*Proof.* It is obvious that Remark 2 does not affect the correctness of Theorem 2 whether or not there are overlapping atomic units in $\mathbb{U}$. However, it is reasonable that the parents of the covered atomic unit must be a subset of or the same as the parents of the covering atomic unit. As illustrated in Figure 8, the atomic unit $\{L_2\}$ is fully covered by $\{L_2, L_3\}$, and both $\{L_2\}$ and $\{L_2, L_3\}$ have parent $\{L_1\}$. After removing $\{L_2\}$ from the active set, the atomic unit $\{L_2, L_3\}$ can be considered as an individual pure child for discovering the new atomic unit $\{L_1\}$ according to Corollary 1. $\qquad\square$

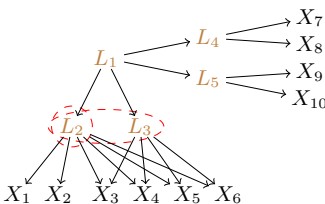

Figure 8: The illustrative examples for remark 2.

### F.6 PROOF OF COROLLARY 1

*Proof.* If one atomic unit $\mathcal{Y}$ is replaceable in the $\mathbb{Y}$ that satisfies Theorem 2 with Remark 2, then we know that in the current causal graph composed of atomic units in $\mathbb{U}$ and their latent confounders, $\mathcal{Y}$ is not directly causally connected to the other atomic units in $\mathbb{Y}\backslash\mathcal{Y}$, except for their common total parent set. The reverse is also true. $\qquad\square$

### F.7 PROOF OF PROPOSITION 2

*Proof.* We demonstrate the utility of Proposition 2 and prove it with the example in Figure 9. With our algorithm, at the beginning, the active set contains only measured variables. After Phase 1 of the first iteration, nothing is identified. During Phase 2 of the first iteration, we cluster the pure children to discover new atomic units. It is obvious that we can cluster $\{X_8\}$ and $\{X_9\}$ to form the atomic unit $\{L_4\}$. However, when clustering for discovering the atomic unit $\{L_1, L_2\}$, the atomic unit $\{X_2\}$ will also be clustered together with any two of $\{\{X_3\}, \{X_4\}, \{X_5\}, \{X_6\}\}$, and considered as a individual pure children of the atomic unit $\{L_1, L_2\}$, though $\{X_2\}$ is actually the individual pure child of $\{L_1\}$. That is because that atomic unit $\{L_1, L_2\}$ covers atomic unit $\{L_1\}$ and when clustering for $\{L_1, L_2\}$, $\{L_1\}$ does not have enough individual pure children to separate it from $\{L_1, L_2\}$. In the following iterations, $\{L_3\}$ will be discovered and identified as a child of $\{L_1, L_2\}$. After that, $\{L_1\}$ can be separated from $\{L_1, L_2\}$ by re-clustering the children of $\{L_1, L_2\}$. We know that $\{X_2\}$ and $\{L_3\}$ can be clustered, which shows the number of parent variables behind them is 1 instead of 2, and therefore, there is a small atomic unit covered by $\{L_1, L_2\}$. $\qquad\square$

### F.8 PROOF OF THEOREM 3

*Proof.* Let $\mathcal{U}_1$ and $\mathcal{U}_2$ be two atomic units. Let $\mathbf{U}_1$ be part of variables in $\text{MS}_b(\mathcal{U}_1)$.

(i) Assume that $\mathcal{U}_1$ and $\mathcal{U}_2$ are two discovered atomic units. $(\text{MS}_a(\mathcal{U}_1 \cup \mathcal{U}_2), \text{MS}_b(\mathcal{U}_1 \cup \mathcal{U}_2))$ can satisfy GIN condition only if $\mathcal{U}_1$ and $\mathcal{U}_2$ have overlapping variables. According to Theorem 5, $\mathcal{S} = \mathcal{U}_1 \cup \mathcal{U}_2$. $Dim(\mathcal{S})$ should be smaller or equal to $Dim(\mathcal{U}_1) + Dim(\mathcal{U}_2) - 1$ so that the graph

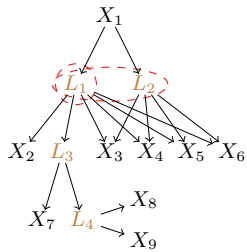

Figure 9: The illustrative examples for proposition 2.

criteria of the GIN condition is satisfied. It means $\mathcal{U}_1$ and $\mathcal{U}_2$ have overlapping variables. When $\mathbf{U}_1$ is the minimal set that $(\mathrm{MS}_a(\mathcal{U}_1 \cup \mathcal{U}_2), \mathbf{U}_1 \cup \mathrm{MS}_b(\mathcal{U}_2))$ follows the GIN condition, we know the true number $\|\mathcal{U}_1 \cup \mathcal{U}_2\|$ is $\|\mathcal{U}_2\| + \|\mathbf{U}_1\| - 1$. Therefore, we know that the two atomic units have $\|\mathcal{U}_1\| - \|\mathbf{U}_1\| + 1$ overlapping variables in total.

(ii) Assume that $\mathcal{U}_1$ and $\mathcal{U}_2$ have $\|\mathcal{U}_1\| - \|\mathbf{U}_1\| + 1$ overlapping variables. According to the graphical criteria of PO-LiNGAM (Theorem 5), we can know that $\mathbf{U}_1$ is the minimal set that $(\mathrm{MS}_a(\mathcal{U}_1 \cup \mathcal{U}_2), \mathbf{U}_1 \cup \mathrm{MS}_b(\mathcal{U}_2))$ follows the GIN condition. □

### F.9 PROOF OF COROLLARY 2

*Proof.* The proof of Corollary 2 is obvious. For a set of atomic units $\mathbb{S}$ that do not overlap with each other and can be covered by atomic unit $\mathcal{U}$, $\|\mathcal{U}\| = \|\mathbb{S}\|$ is equivalent to the fact that each atomic unit in $\mathbb{S}$ is part of $\mathcal{U}$, and together they can form $\mathcal{U}$. □

### F.10 PROOF OF THEOREM 4

*Proof.* In section 3.1, we show that the causal structure among atomic units is identifiable by Theorem 1 and Remark 1. In section 3.2, we show that the new atomic units can be discovered by clustering their pure children with the help of Theorem 2, Remark 2, and Corollary 1. In section 3.3, we refine the discovered atomic units by Theorem 3 and Corollary 2.

The three sections are integrated into our iterative algorithm to discover the causal graph from the leaf to root nodes. For the possible problem with covered small atomic units during the iterative discovery process, we check and identify it through Proposition 2. Furthermore, at the end of discovery process, we check for multiple root atomic units by testing unconditional independence relationships. Therefore, the atomic units, their size, and the causal structure among them can be fully identified with our algorithm. It is obvious that if there are no direct causal relationships between variables within atomic units, the entire causal graph $\mathcal{G}$ is virtually identifiable. □

### F.11 PROOF OF COROLLARY 3

*Proof.* Corollary 3 is similar to Theorem 4, but with stronger assumptions. If each latent variable has at least two pure variable children, each atomic unit we discovered will only contain one variable. Thus, we do not need to consider the problem that the discovered atomic units have overlapping variables. Similarly, we need to check whether there are multiple root atomic units at the end. Roughly speaking, with Theorem 1, Theorem 2, and Corollary 1, the entire causal graph $\mathcal{G}$ is fully identifiable. Furthermore, if there is no latent confounder, we need not consider the discovery of latent variables. The entire causal graph $\mathcal{G}$ is identifiable with only Proposition 1, and the result is the same as the typical LiNGAM discovery algorithm (Shimizu et al., 2006; 2011). □

## G  MORE DETAILS ON SIMULATION EXPERIMENTS

### G.1  DATA GENERATION PROCESS AND IMPLEMENTATION

According to Eq. 1, we generated the causal strength $b_{i,j}$ uniformly from $[-2, -0.5] \cup [0.5, 2]$ and the non-Gaussian noise terms were generated from exponential distributions to the second power.

For BPC and $IL^2H$, Gaussian noise is used. HSIC-based independence tests (Zhang et al., 2018) were used to test the GIN condition.

## G.2 EVALUATION METRICS

We adapted the evaluation metrics from Xie et al. (2022) and Huang et al. (2022) to evaluate the results. They are:

*a.) Correct ordering rate*: the number of correctly inferred causal orderings divided by the total number of inferred causal orderings.

*b.) Error rate in latent variables*: the absolute difference between the inferred number of latent variables and the number of latent variables in the true structure divided by the number of latent variables in the true structure.

*c.) F1-score*: $2\frac{\text{precision}*\text{recall}}{\text{precision}+\text{recall}}$ to measure the similarity between the inferred adjacency matrix and the true adjacency matrix.

## G.3 MORE DETAILS ON F1-SCORE

In this paper, an adjacency matrix is a square matrix whose each element represents whether or not the variables in the graph are connected by edges with direction. For example, $Adj_{\mathcal{G}}(i,j) = 1$ represents there is a direct edge from $i_{th}$ point to $j_{th}$ point, and $Adj_{\mathcal{G}}(i,j) = 0$ represents there is no direct edge from $i_{th}$ point to $j_{th}$ point.

We used the F1-score to calculate the percentage of similarity between estimated and ground-truth adjacency matrices. To calculate the F1-score, we first calculate the precision = $\frac{\text{true positive}}{\text{total test positive}}$ that represents the number of correct inferred edges over number of total inferred edges, and the recall = $\frac{\text{true positive}}{\text{total true positive}}$ that represents the number of correct inferred edges over number of total ground-truth edges. F1-score = $2\frac{\text{precision}*\text{recall}}{\text{precision}+\text{recall}}$, which combines the precision and recall of the adjacency matrices by taking their harmonic mean, and can better represent the structural recovery rate.

In practice, the latent variable indices in the estimated graphs may not match those in the real graphs. To remove this ambiguity, similar to Huang et al. (2022), we permuted the latent variable indices in the estimated graphs and used the one that has the minimal difference from the true graph. In addition, if the estimated number of latent variables is smaller than the true number of latent variables, we add extra latent variables that do not have edges with others to $\mathcal{G}$. If the estimated number of latent variables is larger than the true number of latent variables, we find a subset of the latent variables in $\mathcal{G}$ that best matches the true one.

## G.4 MORE RESULTS OF SIMULATION EXPERIMENTS

Here, we evaluate the running time for four simulation cases. Each case is executed ten times, and the results are averaged. The summarized results are reported in the Table 4.

Table 4: The averaged running times for various cases with different sample sizes.

|  | Case 1 | | | Case 2 | | | Case 3 | | | Case 4 | | |
|---|---|---|---|---|---|---|---|---|---|---|---|---|
|  | 5k | 10k | 50k | 5k | 10k | 50k | 5k | 10k | 50k | 5k | 10k | 50k |
| Time (s) | 8 | 10 | 18 | 25 | 26 | 47 | 18 | 20 | 42 | 2026 | 2164 | 3132 |

To evaluate the performance of our algorithm when some noise is Gaussian-distributed, we modified the first two cases of the simulation experiment. In Case 1, we generate standard Gaussian noise for $X_2$ and $X_4$. In Case 2, we generate standard Gaussian noise for $X_2$ and $X_5$. Each case is executed ten times, and the results are averaged. The summarized results are reported in the Table 5. Although some of the noise is Gaussian-distributed, our method still recovers the true structures.

Table 5: The performance of our method on causal structure graphs in the presence of Gaussian-distributed noise.

| | Case 1 | | | Case 2 | | |
|---|---|---|---|---|---|---|
| | 5k | 10k | 50k | 5k | 10k | 50k |
| Correct Ordering Rate ↑ | 0.93 | 1.0 | 0.94 | 0.91 | 0.95 | 0.95 |
| Error Rate in Latent Variables ↓ | 0.0 | 0.0 | 0.0 | 0.20 | 0.15 | 0.10 |
| F1-score ↑ | 0.95 | 0.97 | 0.95 | 0.91 | 0.93 | 0.96 |

## H  MORE DETAILS ON THE REAL-WORLD DATA EXPERIMENT

The Holzinger and Swineford (1939) dataset consists of mental ability test scores of seventh- and eighth-grade children from two different schools (Pasteur and Grant-White). In the original dataset, there are scores for 26 tests. However, a smaller subset with 9 variables is more widely used in the literature, for example, in Jöreskog (1969). This dataset can be retrieved from R package lavaan (Rosseel, 2012). The variables used in our paper are given in Table 6, which can be broadly categorized into three dimensions: Visual, Textual, and Speeded.

Table 6: The details of 9 variables in 'Holzinger & Swineford 1939' dataset.

| | |
|---|---|
| Visual | Visual perception test from Spearman VPT Part I |
| | Cubes, Simplification of Brighams Spatial Relations Test |
| | Lozenges from Thorndike-Shapes flipped over then identify target |
| Textual | Word Meaning Test |
| | Sentence Completion Test |
| | Paragraph Comprehension Test |
| Speeded | Speeded discrimation of straight and curved caps |
| | Speeded counting of dots in shapes |
| | Speeded addition test |

The second-order factor analysis model dataset is a built-in dataset of the Mplus software (Muthén & Muthén, 2017). The ground truth is the same as in Figure 10(b). Each set of three measured variables has a corresponding first-order latent indicator, all of which are influenced by a second-order latent indicator. A detailed explanation of the dataset and model can be found in `http://www.statmodel.com/HTML_UG/chapter5V8.htm`.

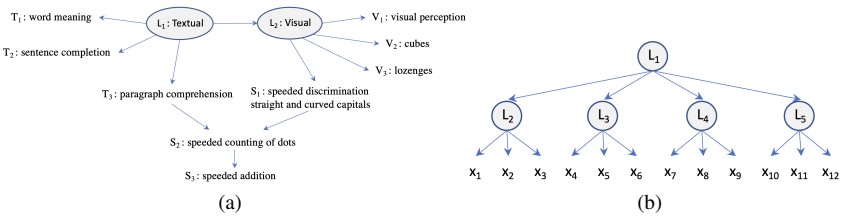

(a)                              (b)

Figure 10: The outputs on (a) 'Holzinger & Swineford 1939' dataset and (b) Second-order factor analysis dataset. The nodes in circles are the latent indicators discovered from measured variables.

## I  NOTATIONS AND TERMS

Table 7: More explanations on Notions and Terms.

| Notation / Term | Description | Initially Used |
|---|---|---|
| X | A measured variable. | First paragraph of Section 2.1 |
| L | A latent variable. | First paragraph of Section 2.1 |
| V | A variable, which is either measured or latent. | First paragraph of Section 2.1 |
| $\mathbf{V}$ | Boldfaced letter; A set of variables, each of which is either measured or latent. | First paragraph of Section 2.1 |
| $\mathcal{V}, \mathcal{U}$ | Script letter; An atomic unit, whose definition is in Definition 3. | Definition 3 |
| $\mathbb{U}, \mathbb{Y}$ | Blackboard bold letter; A set of atomic units. | Theorem 1 |
| $\| * \|$ (e.g., $\mathbf{V}, \mathbb{U}$) | The number of variables in set * ($\mathbf{V}, \mathbf{U}$). | Definition 3 |
| $\| * \|$ (e.g., $\mathbb{U}$) | The number of atomic units in set * ($\mathbb{U}$). | Algorithm 2 |
| $\text{Pa}(*)$ (e.g., V, $\mathcal{U}$) | The parent of * (V, $\mathcal{U}$). | First paragraph of Section 2.1 |
| Pure atomic unit child | Each pure child itself is an atomic unit. | Definition 3 |
| Surrogate variable of $\mathcal{U}$ | Each descendant is a surrogate variable of $\mathcal{U}$. | Definition5 |
| $\text{MS}_{a,b}(\mathcal{U})$ | The measured surrogate variable set a/b of $\mathcal{U}$. | Definition5 |
| Decomposable 'atomic unit' | The discovered 'atomic unit' during phase II, which can be fully decomposed into several small true atomic units during phase III. | First paragraph of Section 3.3 |

