# OpenReview forum: "Structural Estimation of Partially Observed Linear Non-Gaussian Acyclic Model: A Practical Approach with Identifiability"
_ICLR.cc/2024/Conference — ICLR 2024 poster_

### Official Review · Reviewer_XS6k · 2023-10-31

**Soundness:** 4 excellent
**Presentation:** 3 good
**Contribution:** 4 excellent
**Rating:** 8
**Confidence:** 4

**Summary:**

The paper proposed a new causal discovery method for partially observed linear non-Gaussian acyclic model where the structure of observed and latent variables can be quite general. Their algorithm is an iterative one with each iteration consisting of three phases: identifying leafs, identifying atomic units, and refining atomic units. They establish the identifiability of their method under two conditions: (a) any latent variable is in at least one atomic unit and (b) the non-overlapping parts of two overlapping atomic units do not influence each other.

**Strengths:**

1. The causal structure among latent and measured variables are quite general
2. Despite the generality, the causal graph can be identifiable when the two identifiable conditions are met.

**Weaknesses:**

The main weakness I find is the clarity of the writing, which hinders my ability to fully understand the method (hence the current rating). There are a number of places that need to be clarified. Please see my questions below.

One minor weaknesses: the authors used V, boldfaced V, and script V, which can be somewhat hard to read especially the latter two both denote sets. Use a different letter may help. It is particularly confusing in the first line of Definition 3 where script V and boldfaced V are used.

**Questions:**

Q1: How to understand the second bullet point of Definition 3? Does "not partially or fully covered" simply mean intersection is empty?
Q2: Can the authors summarize how their identifiability conditions in relation to existing identifiable conditions for similar models?
Q3: How does condition (a) imply every latent variable must have measured variables as its descendants?
Q4: Definition 5 is a little hard to understand. What is the definition of surrogate variable? How to understand the terms "can be" and "can have" in the definitions? And is the last sentence an assumption? What do you mean by non-redundant set?
Q5: In the last sentence of the first paragraph in Section 3.3, "decomposable atomic units" are not atomic units by definition, right? How "decomposable atomic units" arise from Phases I and II?
Q6: Section 3.4 assume oracle independence test I assume?
Q7: Corollary 3: what "parts" of Phase I&II need to be specified. Additionally, do the authors mean "no latent variables" instead of "no latent confounders" in the last sentence of the corollary?

---

> ### Author Response · Authors · 2023-11-16
> **Rebuttal by Authors (Part 1)**
>
> We thank the reviewer for the insightful comments and valuable feedback. Please see our responses to your concerns point by point below.
>
> **Q1: How to understand the second bullet point of Definition 3? Does "not partially or fully covered" simply mean intersection is empty?**
>
> **A1.1:** Thank you again for your insightful comments. The second bullet of Definition 3 emphasizes that each atomic unit should have a sufficient number of individual pure children to enable its discovery and the inference of causal relations with other atomic units.
>
> **A1.2:** The term "not partially or fully covered" does not simply mean the intersection is empty. In the general case, two overlapping atomic units can hardly be used as individual pure children because of the shared variables. However, **in one special situation where one atomic unit fully covers another, we can use the covering atomic unit as an individual pure child and ignore the covered one.** We introduce this statement to make better use of all available information for discovering higher level latent variables.
>
> For instance, consider atomic unit \{$\mathrm{L}_1$\} with pure children \{\{$\mathrm{X}_1$\},\{$\mathrm{X}_2$\}\} and atomic unit \{$\mathrm{L}_1,\mathrm{L}_2$\} with pure children \{\{$\mathrm{X}_3$\}, \{$\mathrm{X}_4$\}, \{$\mathrm{X}_5$\}, \{$\mathrm{X}_6$\}\}. In this case, atomic unit \{$\mathrm{L}_1$\} is fully covered by atomic unit \{$\mathrm{L}_1,\mathrm{L}_2$\}. When discovering higher-level causal structures (e.g., the parent atomic unit), since they overlap, they cannot be considered as two individual pure children. Here, we should ignore \{$\mathrm{L}_1$\} and then atomic unit \{$\mathrm{L}_1,\mathrm{L}_2$\} can be seen as an individual pure child, as it is not overlapped or covered by other atomic units. Moreover, since it covers \{$\mathrm{L}_1$\}, the parent of \{$\mathrm{L}_1,\mathrm{L}_2$\} must include the parent of \{$\mathrm{L}_1$\}.
>
> After pondering over the presentation, we found that "not partially or fully covered" may be misleading as you point it out. We have now replaced it with "covers all atomic units that share common variables with it" and updated the paper. Thank you for your comment.

---

> ### Author Response · Authors · 2023-11-16
> **Rebuttal by Authors (Part 2)**
>
> **Q2: Can the authors summarize how their identifiability conditions in relation to existing identifiable conditions for similar models?**
>
> **A2:** To the best of our knowledge, our algorithm can identify the most general latent causal graphs, that is, requiring the weakest structural conditions for identifiability. Below is a comparative analysis of the identifiability conditions in several typical methods for latent variables.
>
> |            Structural conditions required           | ours | BPC | FastGIN | LaHME | IL2H |
> |:---------------------------------------------------:|:----:|:---:|:-------:|:-----:|:----:|
> |                      Pure child                     |   ✓  |  ✓  |    ✓    |   ✓   |   ✓  |
> |   Each latent variable having unique pure children  |   ✗  |  ✓  |    ✗    |   ✓   |   ✗  |
> |        No impure child in discovered clusters       |   ✗  |  ✗  |    ✓    |   ✓   |   ✓  |
> |     No shared variable in discovered latent sets    |   ✗  |  -  |    ✓    |   -   |   ✓  |
> |                No latent hierarchies                |   ✗  |  ✓  |    ✓    |   ✗   |   ✗  |
> |         No interaction of measured variables        |   ✗  |  ✗  |    ✓    |   ✓   |   ✓  |
> |     No measured variable causing latent variable    |   ✗  |  ✓  |     ✓   |   ✓   |   ✓  |
> | Markov equivalent outcomes (not fully identifiable) |   ✗  |  ✓  |    ✗    |   ✗   |   ✓  |
>
> Moreover, (i) The number of pure children required in our paper for full identification is smaller or equal to that required by existing practical methods. (ii) Our approach outputs meaningful partial results even if the identifiability condition is violated. Condition(a) is tested during the discovery process, and our algorithm remains robust when Condition(b) is not satisfied.
>
> ---
> **Below, let us explain and discuss the advances of our identifiability condition compared to other methods point by point.**
>
> Specifically, Condition (a) (in the last paragraph of page 3) is intricately linked to the definition of an atomic unit, which in turn is associated with pure children. **The concept of pure children, used inclusively across various methodologies for incorporating latent variables** (Silva et al., 2006; Shimizu et al., 2009; Cai et al., 2019; Xie et al., 2020; 2022; Squires et al., 2022; Huang et al., 2022). Compared to these methods, **the number of pure children required in our paper for the full identification of each atomic unit is smaller or equal, and our method does not require additional structural assumptions as they did**. Meanwhile, **our algorithm precisely identifies pure children and impure children**, while some of the aforementioned methods simply assume that all the children in a discovered cluster are pure. Furthermore, more importantly, **in our algorithm, whether the number of pure children for latent variables satisfies the condition is tested during the discovery process**, which is mainly achieved by Theorem 2, Remark 2, and Corollary 1. Meanwhile, the algorithm halts its inference of the higher-level causal structure when the latent set lacks a sufficient number of pure children. As a consequence, **even if some latent variables do not have enough pure children, instead of output spurious latent variables, the atomic units, whose causal orders are lower or equal to unidentified latent variables, and the causal relations among them are still identifiable**. Please refer to **Appendix B** for a detailed explanation illustrating the consequences if condition(a) is violated.
>
> Condition(b) is often overlooked by other methods, which simply assume different discovered latent sets do not share common latent variables. **Our paper thoroughly considers the potential situations of latent atomic units**, which are established by Remark 1, Remark 2, Proposition 2, Theorem 3, and Corollary 2. Importantly, similar to Condition(a), **our algorithm still identifies the causal graph even if Condition(b) is violated**. **In our algorithm, we actually neglected the causal relations from their non-overlapping parts, regardless of whether their non-overlapping parts influence each other**, since they are indeterminable. This is addressed by Remark 1 and Remark 2 in the paper. Please also refer to **Appendix B** for a detailed explanation with graphs.
>
> ---
>
> Reference:
>
> [1] Silva, R., Scheines, R., Glymour, C., Spirtes, P., & Chickering, D. M. (2006). Learning the Structure of Linear Latent Variable Models. Journal of Machine Learning Research, 7(2).
>
> [2] Shimizu, S., Hoyer, P. O., & Hyvärinen, A. (2009). Estimation of linear non-Gaussian acyclic models for latent factors. Neurocomputing, 72(7-9), 2024-2027.
>
> [3] Cai, R., Xie, F., Glymour, C., Hao, Z., & Zhang, K. (2019). Triad constraints for learning causal structure of latent variables. Advances in neural information processing systems, 32.

---

> ### Author Response · Authors · 2023-11-16
> **Rebuttal by Authors (Part 3)**
>
> **Additional reference for Q2:**
>
> [4] Xie, F., Cai, R., Huang, B., Glymour, C., Hao, Z., & Zhang, K. (2020). Generalized independent noise condition for estimating latent variable causal graphs. Advances in neural information processing systems, 33, 14891-14902.
>
> [5] Xie, F., Huang, B., Chen, Z., He, Y., Geng, Z., & Zhang, K. (2022, June). Identification of linear non-gaussian latent hierarchical structure. In International Conference on Machine Learning (pp. 24370-24387). PMLR.
>
> [6] Squires, C., Yun, A., Nichani, E., Agrawal, R., & Uhler, C. (2022, June). Causal structure discovery between clusters of nodes induced by latent factors. In Conference on Causal Learning and Reasoning (pp. 669-687). PMLR.
>
> [7] Huang, B., Low, C. J. H., Xie, F., Glymour, C., & Zhang, K. (2022). Latent hierarchical causal structure discovery with rank constraints. Advances in Neural Information Processing Systems, 35, 5549-5561.
>
>
> **Q3: How does condition (a) imply every latent variable must have measured variables as its descendants?**
>
> **A3:** Condition (a) states that any latent variable is in at least one atomic unit. According to Definition 3, each atomic unit is either a measured variable or a variable set involving latent variables and having a sufficient number of pure children. Consequently, leaf nodes of the causal graph must be measured variables. Therefore, every latent variable must have measured variables as its descendants.
>
> **Q4: What is the definition of surrogate variable? How to understand the terms "can be" and "can have" in the definitions? And is the last sentence an assumption? What do you mean by non-redundant set?**
>
> **A4.1:** The definition of surrogate variable is referred to Definition 5. Let $\mathcal{P}$ be an atomic unit involving latent variables and atomic unit $\mathcal{C} \in \mathrm{PCh}(\mathcal{P})$. Surrogate variables of $\mathcal{P}$ are the variables in each $\mathcal{C}$ or up to $\|\mathcal{C}\|$ surrogate variables of $\mathcal{C}$. It is defined recursively and used to obtain information about latent atomic units from their measured descendants. In light of your comment. we have now modified Definition 5 and provided the definition of surrogate variable explicitly.
>
> **A4.2:** The term "can be" in the definition indicates that there always exist descendants to serve as surrogate variables for each atomic unit. For the same reason, when we state "each atomic unit can have measured variables as its surrogate variables", there always exist measured descendants to serve as surrogate variables for each atomic unit. To avoid misunderstanding, we have now modified the description in Definition 5.
>
> **A4.3:** The last sentence is not an assumption but a natural consequence. In Definition 3, the atomic unit involving latent variables is defined as a set having a sufficient number of pure children. Therefore, each atomic unit is naturally having at least two separate measured surrogate variable sets. In the discovery process, it is achieved because each atomic unit is discovered from a sufficient number of measured variables (leaves of the causal graph, by Theorem 2 and Remark 2). This statement merely describes the characteristics of the measured surrogate set. We have now modified the description in the Definition 5.
>
> **A4.4:** "non-redundant set" $\mathcal{P}$ means that $\mathcal{P}$ is the minimal set that d-separates its two separate measured surrogate sets. In light of your question, we have now replaced the phrase "non-redundant" with "minimal" in Definition 5.

---

> ### Author Response · Authors · 2023-11-16
> **Rebuttal by Authors (Part 4)**
>
> **Q5: In the last sentence of the first paragraph in Section 3.3, "decomposable atomic units" are not atomic units by definition, right? How "decomposable atomic units" arise from Phases I and II?**
>
> **A5:** Thank you for your insightful comments. Indeed, **decomposable 'atomic units' are not atomic units by definition.** We used the quotation marks in decomposable 'atomic units' to indicate that they are not actually atomic units but we failed to distinguish them from normal atomic units in Phase I and II, as noted in footnote 4. In our approach, **when we discover new atomic unit candidates by identifying and clustering their pure children during Phase II, we are initially unaware of whether they are decomposable 'atomic units'. It is in Phase III of each iteration that we examine the overlapping of newly discovered atomic units using Theorem 3, and then verify if any of them is decomposable through Corollary 2.** In light of your question, we have now updated the description in Section 3.3. Below is an example to help illustrate the concept.
>
> For instance, as stated in Example 9 and 12, consider Figure 1, where \{$\mathrm{L}_1, \mathrm{X}_2$\} shares common pure children \{\{$\mathrm{X}_5$\}, \{$\mathrm{X}_6$\}, \{$\mathrm{X}_7$\}, \{$\mathrm{X}_8$\}\} and, simultaneously, \{$\mathrm{L}_1$\} has its pure children \{\{$\mathrm{X}_3$\},\{$\mathrm{X}_4$\}\}. After completing Phases I and II, three atomic units exist in the active atomic unit set: \{$\mathrm{L}_1$\}, \{$\mathrm{L}_1, \mathrm{X}_2$\}, and \{$\mathrm{X}_2$\}. During phase III, we find that \{$\mathrm{L}_1, \mathrm{X}_2$\} can be decomposed by \{$\mathrm{L}_1$\} and \{$\mathrm{X}_2$\} by Theorem 3 and Corollary 2.
>
> **Q6: Section 3.4 assume oracle independence test I assume?**
>
> **A6:** Yes, you are totally correct. For establishing the theoretical identifiability result (which are asymptotic results), we assume oracle GIN tests.
>
> **Q7: Corollary 3: what "parts" of Phase I and II need to be specified. Additionally, do the authors mean "no latent variables" instead of "no latent confounders" in the last sentence of the corollary?**
>
> **A7.1:** Thank you very much for the effort you put in reviewing the paper. In Corollary 3, given the scenario that each latent variable has at least two pure variable children, we exclude the consideration of overlapping problems among discovered latent atomic units. This is because each atomic unit contains only one variable, whether latent or measured, eliminating the need for Remark 1, Remark 2, Proposition 2, and Phase III. **The entire causal graph can be identified only through Theorem 1, Theorem 2, and Corollary 1.** This is why we called it "parts" of Phase I and II. We have now revised Corollary 3 in the updated version to make it explicit.
>
> **A7.2:** The terms "latent confounders" and "latent variables" are used interchangeably in this paper and in other classical causal discovery works, such as Spirtes et al.(2013).
>
> Reference:
> [1] Spirtes, P. L., Meek, C., & Richardson, T. S. (2013). Causal inference in the presence of latent variables and selection bias. arXiv preprint arXiv:1302.4983.
>
> **Q8: The authors used V, boldfaced V, and script V, which can be somewhat hard to read especially the latter two both denote sets. Use a different letter may help.**
>
> **A8:** Thank you for your kind suggestion. We used different letters V, L, and X for general variables, latent variables, and measured variables, respectively. Meanwhile, as you point out, both general variable set and atomic unit are variable sets, even if they have distinct meanings. After reflecting on it, it seems sensible to stick with such notations in order to make it clear that
> $\mathrm{V}_i$ is a element in both $\mathbf{V}$ and $\mathcal{V}$. Introducing additional letters here might compromise the clear correspondence between the set and the variables within it.
>
> For better clarity, we have added corresponding explanations beside each notation when it is introduced for the first time. Specifically, it can be seen from the first line of Definition 3 that $\mathcal{V}$ denotes an atomic unit and $\mathbf{V}$ denotes a general variable set. Thank you once again for your suggestion. If you feel it is not clear enough and should be updated to improve the clarity, please kindly let us know and we will be grateful and happy to make changes accordingly.
>
> **Q9: Clarity of the writing.**
>
> **A9:** Thank you for your time dedicated to reviewing our paper and your valuable comments and questions. **We have now updated our paper accordingly and highlighted the changes in blue based on your comments.** We hope your concerns are properly addressed. Please kindly share any additional questions and we will respond immediately.
>
> Once again, we appreciate your review.

---

> > ### Comment · Reviewer_XS6k · 2023-11-16
> > **Rating adjusted**
> >
> > I'd like to thank the authors' detailed responses. I have adjusted my score as a result. Overall, the paper is very good.

---

> > > ### Author Response · Authors · 2023-11-17
> > >
> > > We sincerely appreciate your prompt response and positive feedback on our paper. Thank you!

---

### Official Review · Reviewer_53sg · 2023-11-01

**Soundness:** 3 good
**Presentation:** 3 good
**Contribution:** 3 good
**Rating:** 6
**Confidence:** 3

**Summary:**

The problem of causal structure learning with latent confounders is considered, with linearity and non-Gaussianity assumptions, and an additional assumption relating to notions of atomic units and their pure children in the graph. A completeness theorem is given showing structures can be learned up to atomic units and experiments on both generated and real data are done.

**Strengths:**

The work is well-situated within the causal structure learning literature, where various classes of assumptions are studied. In this context the additional graphical assumptions that they consider are novel and interesting. As far as I can tell, the results are sound and the experimental results are demonstrative.

**Weaknesses:**

The presentation is somewhat dense. See questions below.

**Questions:**

Can the authors give real examples of when the assumptions relating to atomic units (PO-LiNGAM identifiability condition, including part b) would be reasonable to assume, and when they wouldn't? This would help me get a better grasp of the significance. As it stands we have these assumptions that are clearly stated but I don't know when they can really be made.

Minor comments:
- Abstract: "fragile" -> "sensitive"

---

> ### Author Response · Authors · 2023-11-16
> **Rebuttal by Authors**
>
> We thank the reviewer for the insightful comments and valuable feedback. Please see our responses to your concerns below.
>
> **Q1: Can the authors give real examples of when the assumptions relating to atomic units (PO-LiNGAM identifiability condition, including part b) would be reasonable to assume, and when they wouldn't?**
>
> **A1:** Thank you for your thoughtful consideration. **Our method is applicable across a wide range of fields, including psychology, education, business, brain signals, genetic data, and various social and health sciences.**
> - As you can see in our experiment on one valuable public mental dataset (Holzinger & Swineford, 1939), many external phenotypes are separately affected by the common latent factors that are related to each other, which is also validated in Cui et al. (2019).
> - Meanwhile, in the education field, for instance, Barbara Byrne conducted a study to investigate the impact of organizational and personality factors on burnout in full-time elementary teachers and found that many measured variables are independently affected by some common latent factors (Byrne, 2010). We have analyzed the dataset and the results are consistent with the previous finding, hence we skipped it in the paper.
> - There may be other problems in brain science (fMRI signals) and gene regulatory networks as well (Power et al.,2014; Squires et al., 2022). Handling such large datasets is one of our future work.
>
> To the best of our knowledge, **our algorithm can identify the most general latent causal graphs, that is, requiring the weakest structural conditions for identifiability.** The number of pure children required in condition(a) for full identification is smaller or equal to that required by existing practical methods (e.g., Xie et al., 2020; Huang et al., 2022). **Condition(b) is not a new condition. Instead, it is often overlooked and implicitly exists in other methods (Xie et al., 2020; Huang et al., 2022).**
>
> Furthermore, we would like to add that **whether the number of pure children for latent variables satisfies condition(a) is tested during our discovery process**, which is mainly achieved by Theorem 2, Remark 2, and Corollary 1. Because our algorithm sequentially identifies the whole causal graph from leaves to roots and the algorithm halts its inference of the higher-level causal structure when the latent set lacks a sufficient number of pure children, it does not output spurious latent variables. As a consequence, **even if some latent variables do not have enough pure children, the atomic units, whose causal orders are lower or equal to unidentified latent variables, and the causal relations among them are still identifiable**. Moreover, our algorithm explicitly neglects causal relations from non-overlapping parts that cannot be uniquely identified, as demonstrated by Remark 1 and Remark 2. So, **the algorithm still works even in cases where condition(b) is not met**. A detailed explanation with graphs can be found in Appendix B of our paper. In light of your comments, we have now also added some experiments in which two structural conditions are violated in Appendix B2. Thus, a wide range of data can be attempted with our algorithm, even if we have no prior knowledge of the data.
>
>
> Reference:
>
> [1] Holzinger, K. J., & Swineford, F. (1939). A study in factor analysis: The stability of a bi-factor solution. Supplementary educational monographs.
>
> [2] Cui, R., Bucur, I. G., Groot, P., & Heskes, T. (2019). A novel Bayesian approach for latent variable modeling from mixed data with missing values. Statistics and Computing.
>
> [3] Byrne, B. M. (2010). Structural equation modeling with AMOS: Basic concepts, applications, and programming (2nd ed.).
>
> [4] Power, J. D., Schlaggar, B. L., & Petersen, S. E. (2014). Studying brain organization via spontaneous fMRI signal. Neuron, 84(4), 681-696.
>
> [5] Squires, C., Yun, A., , et al. (2022, June). Causal structure discovery between clusters of nodes induced by latent factors. CLeaR.
>
> [6] Xie F, Cai R, Huang B, et al. (2020). Generalized independent noise condition for estimating latent variable causal graphs. NeurIPS.
>
> [7] Huang, B., Low, C. J. H., Xie, F., Glymour, C., & Zhang, K. (2022). Latent hierarchical causal structure discovery with rank constraints. NeurIPS.
>
> **Q2: Minor comments: Abstract: "fragile" -> "sensitive"?**
>
> **A2:** Thank you for the suggestion and we have now updated it accordingly.
>
> **Q3: The presentation is somewhat dense.**
>
> **A3:** We appreciate the effort you put in reviewing the paper. We tried to improve the paper's readability by incorporating examples for each definition and theorem, along with transition segments between them. At the same time, given its technical nature, it seems that we should try to include all the essential information in the main body, which may make it look dense.
>
> Thank you again for your valuable feedback and hope that all your concerns are properly addressed.

---

> ### Author Response · Authors · 2023-11-20
> **Could you kindly check whether our response and revision properly addressed your concern?**
>
> Dear Reviewer 53sg,
>
> We appreciate your time devoted to reviewing this paper. It would be further highly appreciated if you let us know whether our response and the change in the paper properly addressed your concerns. Thank you very much!
>
> With best regards,
> Authors of submission 1403

---

> > ### Comment · Reviewer_53sg · 2023-11-23
> >
> > Thanks for the response. I maintain my current rating.

---

> > > ### Author Response · Authors · 2023-11-23
> > > **Please kindly let us know if you have any additional concerns or questions**
> > >
> > > Dear Reviewer 53sg,
> > >
> > > Thank you very much for reviewing our response. If you have any additional concerns or questions, please let us know. We would be grateful and happy to respond accordingly.
> > >
> > > Best regards,
> > > Authors of submission 1403

---

### Official Review · Reviewer_e967 · 2023-11-01

**Soundness:** 3 good
**Presentation:** 4 excellent
**Contribution:** 3 good
**Rating:** 6
**Confidence:** 3

**Summary:**

In the paper, the authors suggest a general version of LiNGAM, called PO-LiNGAM, where the existence of latent variables is permitted. While previous Linear non-Gaussian variants assume either no-latent variable, specific position or the number of latent variables , the proposed method relaxes such assumptions. When reconstructing a causal graph, the authors use a set of variables called the atomic unit, which represents the basic building block of casual structures. The PO-LiNGAM identification procedure consists of three step: (Phase 1) : Identify a leaf node and its parents; (Phase 2): Discover atomic units; and (Phase 3) : Refine the atomic units. Discovering relationships between atomic units is based on theorems under GIN (Generalized Independent Noise) condition which is an existing graphical condition for identifying a latent hierarchical structure.

**Strengths:**

In order to alleviate the strong assumptions of existing linear non-gaussian models, the authors cluster the variables into atomic units and obtain the information of latent variables from their measured descendants, called the measurement surrogate variable set. In other words, the authors establish the theorems, which view the existing GIN condition from the perspective of the atomic unit. This approach actively utilizes linear transitivity to infer latent variables. The author clearly describes the entire process of the method, and helps the reader's understanding by showing various examples. A sufficient amount of experiments demonstrates the validity of the paper.

**Weaknesses:**

The contribution of the model PO-LiNGAM is that it is free from assumptions about latent variables compared to existing models. However, for the algorithm to work properly, each atomic unit must have a sufficient number of pure atomic unit children. It is difficult to say that this is not a strong condition. The authors are aware of this issue and presented relaxing this as future work. One thing to add is that the only explanation of notations in the paper is provided through table 1 in figure 1. It is difficult to clearly understand the notations just from the table. For example, in definition 3, the term 'pure atomic unit child' is used for the first time. However, no clear explanation for this term is provided. It would be better if more specific notation explanations are added.

**Questions:**

there is a minor typo: In Remark 1, 2nd sentence, we remove them form → we remove them from

---

> ### Author Response · Authors · 2023-11-16
> **Rebuttal by Authors (Part 1)**
>
> We thank the reviewer for the insightful comments and valuable feedback. Please see our responses to your concerns point-by-point below.
>
> **Q1: For the algorithm to work properly, each atomic unit must have a sufficient number of pure atomic unit children. It is difficult to say that this is not a strong condition.**
>
> **A1:** Thank you for your valuable comment and the opportunity to highlight the contribution of our work. **To the best of our knowledge, our algorithm can identify the most general latent causal graphs among all existing practical methods (e.g., Shimizu et al., 2009; Cai et al., 2019; Xie et al., 2020; 2022; Squires et al., 2022; Huang et al., 2022).** That is, among them, it requires the weakest structural conditions for identifiability.
>
> Indeed, in some real cases, the identifiability condition may not hold. However, as illustrated in Appendix B of the paper, **our algorithm can still discover meaningful partial causal graphs even if the structural conditions are violated**, which is another important advance. In light of your comments, to exhibit the robustness of the algorithm, we have now conducted some experiments in which two structural conditions are violated. They are (1) some latent sets do not have sufficient pure children, and (2) for two overlapping atomic units, the non-overlapping part of one affects that of another. The results have now been included in Appendix B2 of updated version.
>
> ---
> **Below, let us explain and discuss the identifiability condition and algorithm robustness in more detail.**
>
> First of all, it is worth mentioning that for the atomic unit consisting of measured variables alone, no pure child is needed. For the atomic unit involving latent variables, **pure children are needed with the following two reasons.** The first reason is to **ensure that the discovered latent atomic unit is not redundant.** For instance, if latent variables $\mathrm{X}_1 \rightarrow \mathrm{L}_1 \rightarrow \mathrm{X}_2$ and $\mathrm{L}_1$ only have one child $\mathrm{X}_2$, then $\mathrm{L}_1$ is redundant and can be eliminated. The second reason is that we considered an inclusive scenario where any causal relations may occur without prior knowledge of the presence and positioning of latent variables. Pure children become **crucial for fully identifying the causal relations among latent atomic units,** as established in Theorem 1.
>
> The concept of pure children has been widely employed across various methods for incorporating latent variables (Shimizu et al., 2009; Cai et al., 2019; Xie et al., 2020; 2022; Squires et al., 2022; Huang et al., 2022). **In comparison to these approaches, our algorithm necessitates a smaller or equal number of pure children for full identification**. Meanwhile, **we do not need additional structural assumptions as they required**, such as measured variables not being ancestors of latent variables and no direct causal relations among measured variables. Moreover, in Xie et al. (2020) and Huang et al. (2022), they took the easy way out by simply assuming that all variables in a discovered cluster are pure. In our algorithm, **we precisely differentiate pure children from impure children**.
>
> Another important advance made in this contribution is that **our algorithm can still discover meaningful partial causal graphs even when the identifiability condition is not satisfied**. Please notice that our algorithm sequentially identifies the whole causal graph from leaves to roots; moreover, **whether the number of pure children for latent variables satisfies the condition is tested during the discovery process**, which is mainly achieved by Theorem 2, Remark 2, and Corollary 1. As a consequence, the algorithm halts its inference of the higher-level causal structure when the latent set lacks a sufficient number of pure children so that it does not output wrong claims about the existence of latent variables. In other words, since this condition is violated, it can not detect all latent variables, but we can still trust the discovered ones. **The atomic units, whose causal orders are lower or equal to the unidentified latent set, and the causal relations among them are still identifiable**.
>
>  Moreover, **it is also acceptable if the second term of the identifiability condition (where the non-overlapping parts of two overlapping atomic units do not influence each other) is not satisfied**, since our algorithm explicitly neglects causal relations from non-overlapping parts that cannot be uniquely identified, as demonstrated by Remark 1 and Remark 2. A detailed explanation with graphs can be found in Appendix B of our paper.
>
> ---
>
> Reference:
>
> [1] Shimizu, S., Hoyer, P. O., & Hyvärinen, A. (2009). Estimation of linear non-Gaussian acyclic models for latent factors. Neurocomputing, 72(7-9), 2024-2027.

---

> ### Author Response · Authors · 2023-11-16
> **Rebuttal by Authors (Part 2)**
>
> **Additional reference for Q1:**
>
> [2] Cai, R., Xie, F., Glymour, C., Hao, Z., & Zhang, K. (2019). Triad constraints for learning causal structure of latent variables. Advances in neural information processing systems, 32.
>
> [3] Xie, F., Cai, R., Huang, B., Glymour, C., Hao, Z., & Zhang, K. (2020). Generalized independent noise condition for estimating latent variable causal graphs. Advances in neural information processing systems, 33, 14891-14902.
>
> [4] Xie, F., Huang, B., Chen, Z., He, Y., Geng, Z., & Zhang, K. (2022, June). Identification of linear non-gaussian latent hierarchical structure. In International Conference on Machine Learning (pp. 24370-24387). PMLR.
>
> [5] Squires, C., Yun, A., Nichani, E., Agrawal, R., & Uhler, C. (2022, June). Causal structure discovery between clusters of nodes induced by latent factors. In Conference on Causal Learning and Reasoning (pp. 669-687). PMLR.
>
> [6] Huang, B., Low, C. J. H., Xie, F., Glymour, C., & Zhang, K. (2022). Latent hierarchical causal structure discovery with rank constraints. Advances in Neural Information Processing Systems, 35, 5549-5561.
>
> **Q2: It is difficult to clearly understand the notations just from the table. For example, in definition 3, the term 'pure atomic unit child' is used for the first time.**
>
> **A2:** We appreciate your time dedicated to reviewing our paper and your suggestion to further improve our paper.
>
> (a) Based on your suggestion, we now have provided a more comprehensive table in Appendix I, which explains the notations and terms with more details and indicates where they were initially introduced. Moreover, we ensured that all terms are explained the first time they appear.
>
> (b) The term 'pure atomic unit child' indicates that the pure child itself is an atomic unit. We used this term following the definition of 'pure child' and 'atomic unit' and have now incorporated this explanation in the footnote on page 3 to make it clearer.
>
> **Q3: minor typo: In Remark 1, 2nd sentence, we remove them form → we remove them from.**
>
> **A3:** Thank you for spotting out this typo and we have now fixed it.
>
> Thank you again for your valuable feedback and hope that all your concerns are properly addressed.

---

> ### Author Response · Authors · 2023-11-20
> **Could you please let us know whether our response and revision properly addressed your concern?**
>
> Dear Reviewer e967,
>
> Thank you very much for your time spent on our submission and your questions. We have tried to address your concerns in the response and updated submission--any feedback from you would be appreciated. If you have further comments, please kindly let us know--we hope for the opportunity to respond to them.
>
> With best regards,
> Authors of submission 1403

---

> > ### Comment · Reviewer_e967 · 2023-11-21
> > **comment by reviewer**
> >
> > I have carefully read the authors' responses. I appreciate your kind responses. I confirm that the feedbacks are incorporated into the paper, and I make some slight adjustments to the scores.

---

> > > ### Author Response · Authors · 2023-11-21
> > > **We appreciate your timely feedback**
> > >
> > > Dear Reviewer e967,
> > >
> > > Thank you very much for examining our response and confirming our feedback. Please kindly let us know if you have other concerns or questions. We will be grateful and happy to response accordingly. In case that all your concerns are properly addressed, could you please consider updating the rating score to reflect it? Your feedback is highly appreciated.
> > >
> > > Best wishes,
> > > Authors of submission 1403

---

### Official Review · Reviewer_BTCR · 2023-11-01

**Soundness:** 3 good
**Presentation:** 3 good
**Contribution:** 3 good
**Rating:** 6
**Confidence:** 4

**Summary:**

This paper addresses the challenging issue of causal discovery in the presence of latent variables. It considers a general scenario in which both measured and latent variables collectively form a partially observed causally sufficient linear system, allowing latent variables to be situated anywhere within the causal structure. The paper establishes a theoretical foundation, demonstrating that with the application of high-order statistics, the causal graph can be (almost) fully identified. This identification is achieved under the condition that each latent set contains a sufficient number of pure children, whether they are latent or measured variables. Importantly, this extends beyond LiNGAM, a model without latent variables.

Expanding on the identification theorem, the paper introduces a systematic algorithm for identifying the causal graph. This algorithm involves three iteratively executed phases that sequentially identify latent variables and infer causal relationships among both latent and measured variables. Experimental results validate the effectiveness of the proposed method in accurately recovering the causal structure, even when latent variables are influenced by measured variables.

**Strengths:**

They have introduced an algorithm that extends the LiNGAM algorithm to accommodate latent confounder conditions. Moreover, they do not impose any constraints on the presence or positioning of latent variables. Additionally, their algorithm has the capability to handle overlapping variables within the discovered latent sets.

Clarity: The paper is well-organized, with examples that effectively demonstrate how the algorithm functions.

**Weaknesses:**

The paper  appears to have no discernible weaknesses. It seems to be well-structured and comprehensive, with ample supporting materials in the appendix. The methodology and presentation seem robust.

Given that I am not an expert in this domain, I might leave the 'weaknesses' section blank for the time being and await feedback from other reviewers during the rebuttal phase.

**Questions:**

1. The decomposition of atomic units among latent variables is not unique, correct? If so, how can this be proven? If not, could this potentially impact the final results?

2. What is the computational time required for this algorithm?

---

> ### Author Response · Authors · 2023-11-16
> **Rebuttal by Authors**
>
> We thank the reviewer for the insightful comments and valuable feedback. Please see our responses to your questions point by point below.
>
> **Q1: The decomposition of atomic units among latent variables is not unique, correct? If so, how can this be proven? If not, could this potentially impact the final results?**
>
> **A1:** Thank you for allowing us to clarify this point. **The decomposition of each atomic unit is unique**. Each atomic unit is a unique minimal component of the causal graph by definition and can be identified by leveraging the GIN test with the search procedure given in Algorithm 1.
>
> Specifically, **for any atomic units involving latent variables, each of them is determined by its unique set of pure children, as outlined in the definition of an atomic unit (Definition 3)**. For example, suppose that \{$\mathrm{L}_1$\} has pure children \{\{$\mathrm{X}_1$\},\{$\mathrm{X}_2$\}\}, and that \{$\mathrm{L}_1$,$\mathrm{L}_2$\} has pure children \{\{$\mathrm{X}_3$\}, \{$\mathrm{X}_4$\}, \{$\mathrm{X}_5$\}, \{$\mathrm{X}_6$\}\}. In this case, \{$\mathrm{L}_1$\} and \{$\mathrm{L}_1$,$\mathrm{L}_2$\} are two atomic units uniquely determined by their respective pure children, even though they share $\mathrm{L}_1$. In our algorithm, we identify each new atomic unit candidate in Phase II by identifying and clustering its pure children, with the help of Theorem 2, Remark 2, and Corollary 1. Following that, in Phase III, we further identify whether any two discovered atomic units share common variables with the help of Theorem 3, which is relatively under-explored in literature.
>
> Moreover, **the third bullet of Definition 3 specifies that each atomic unit is a minimal set that cannot be decomposed into smaller atomic units**. For instance, suppose that \{$\mathrm{L}_2$\} also has its own pure children \{\{$\mathrm{X}_7$\},\{$\mathrm{X}_8$\}\}, then \{$\mathrm{L}_1$\} and \{$\mathrm{L}_2$\} are two atomic units, but \{$\mathrm{L}_1,\mathrm{L}_2$\} is not, as \{$\mathrm{L}_1,\mathrm{L}_2$\}  is not minimal and can be decomposed into two smaller atomic units, namely, \{$\mathrm{L}_1$\} and \{$\mathrm{L}_2$\} instead. The refinement process is based on Theorem 3 and Corollary 2 in Phase III of each iteration.
>
> **Q2: What is the computational time required for this algorithm?**
>
> **A2:** Thank you for your comprehensive consideration. As mentioned in the line before Theorem 4 in Section 3.4, computational complexity analysis was given in Appendix E. Running time of some of the experiments was also provided in Appendix G4. In light of your question, we have now incorporated a summary of the computational complexity to the main text as well.
>
> Specifically, **the computational complexity is contingent upon the number of variables (including both latent and measured variables) and the density of the underlying causal graph**. These factors determine the number of iterations needed to discover the entire graph. In Appendix E, we presented the upper bounds of computational complexity for each phase. Denote by $n$ the number of active atomic units at the beginning of each phase. **The complexity of Phase I is upper-bounded by $\mathcal{O}(n\sum^{n-1}_ {k=1}\binom{n-1}{k})$.** **The worst-case complexity for Phase II is $\mathcal{O}(\sum^{n}_{k=2}\binom{n}{k})$.** Denote by $l$ the size of the largest atomic unit in the active atomic unit set. **The complexity of Phase III is upper-bounded by $\mathcal{O}(l\binom{n}{2})$.**
>
> In Appendix G4, we presented the average running time for four simulation cases, acquired using a standard personal computer. Roughly speaking, **for simple causal graphs (e.g., Case 1, 2, 3), the algorithm took from a few seconds to tens of seconds. For a more complex causal graph (e.g., Case 4), it required additional time to run more iterations.** The time differences among these cases may provide insights into how the complexity of the causal graph influences computation time.
>
> Thank you again for the time you dedicated to reviewing the paper and hope that all your concerns are properly addressed.

---

> ### Author Response · Authors · 2023-11-20
> **Could you please let us know whether our response and revision properly addressed your concern?**
>
> Dear Reviewer BTCR,
>
> Thanks for your time dedicated to reviewing this paper. It would be further highly appreciated if you let us know whether our response and the change in the paper properly addressed your concerns. Thanks a lot!
>
> With best regards,
> Authors of submission 1403

---

### Meta-Review · Area_Chair_jegc · 2023-12-06

**Metareview:**

The paper provides unifying principles for learning the structure of causal linear models, particularly those heavily confounded by latent variables that can be explicitly discovered as a part of the process. Reviewers in general agreed that the contribution provides meaningful advances to this problem that practitioners will appreciate.

**Justification For Why Not Higher Score:**

This is not an easy paper to digest, and although results are welcome they still build heavily on the existing literature.

**Justification For Why Not Lower Score:**

The generalizations introduced in this paper alleviate the concern of many applications where the usual assumptions may be too stringent to be satisfied.

---

### Decision · Program_Chairs · 2024-01-16

Accept (poster)